# PUFFFIN: an ultra-bright, customisable, single-plasmid system for labelling cell neighbourhoods

Tamina Lebek [ID][1], Mattias Malaguti [ID][1,7], Giulia LM Boezio [ID][2], Lida Zoupi [ID][3,4], James Briscoe [ID][2], Alistair Elfick [ID][5,6] & Sally Lowell [ID][1][✉]

## Abstract

**Cell communication coordinates developmental processes, maintains homeostasis, and contributes to disease. Therefore, understanding the relationship between cells in a shared environment is crucial. Here we introduce Positive Ultra-bright Fluorescent Fusion For Identifying Neighbours (PUFFFIN), a cell neighbour-labelling system based upon secretion and uptake of positively supercharged fluorescent protein s36GFP. We fused s36GFP to mNeonGreen or to a HaloTag, facilitating ultra-bright, sensitive, colour-of-choice labelling. Secretor cells transfer PUFFFIN to neighbours while retaining nuclear mCherry, making identification, isolation, and investigation of live neighbours straightforward. PUFFFIN can be delivered to cells, tissues, or embryos on a customisable single-plasmid construct composed of interchangeable components with the option to incorporate any transgene. This versatility enables the manipulation of cell properties, while simultaneously labelling surrounding cells, in cell culture or in vivo. We use PUFFFIN to ask whether pluripotent cells adjust the pace of differentiation to synchronise with their neighbours during exit from naïve pluripotency. PUFFFIN offers a simple, sensitive, customisable approach to profile non-cell-autonomous responses to natural or induced changes in cell identity or behaviour.**

**Keywords** Neighbour Labelling; Synthetic Signalling; PUFFFIN; HaloTag; Pluripotency
**Subject Categories** Biotechnology & Synthetic Biology; Stem Cells & Regenerative Medicine; Methods & Resources

## Introduction

All multicellular life depends on cell-cell communication. It is, therefore, important to understand the mechanisms by which cells influence their neighbours. Traditional approaches to characterise neighbour responses are often based on image analysis, which is limited by the available combination of fluorescence channels (Lin et al, 2015; Lun and Bodenmiller, 2020). This limitation is being overcome by advances in spatial transcriptomics technologies (Walker et al, 2022), but these are technically challenging for analysing complex tissues at single-cell resolution, remain beyond the technical and financial reach of many laboratories, and do not allow for functional assays to test the properties of live neighbours. Alternative approaches are needed to facilitate simple, reliable, sensitive, unbiased analysis of neighbour responses.

One such approach is to engineer synthetic signalling pathways that allow particular cells to induce expression of fluorescent proteins in neighbours, allowing those neighbours to be isolated by fluorescence-activated cell sorting (FACS) and then profiled, for example, using omics analysis, or subjected to functional assays to assess fate, potency or other behaviours. Recent advances in protein design and cell engineering provide a range of such neighbour-labelling systems (Malaguti et al, 2024) with a notable example being synNotch (Gordon et al, 2015; Morsut et al, 2016; Roybal et al, 2016; Huang et al, 2016) and its derivatives, such as synNQ (He et al, 2017), TRACT (Huang et al, 2017), SyNPL (Malaguti et al, 2022), gLCCC (Zhang et al, 2022) or diffusible synNotch (Toda et al, 2020). Other synthetic signalling systems include Tango (Talay et al, 2017; Sorkaç et al, 2023) and its light-inducible versions (Kim et al, 2017, 2019; Cho et al, 2022), G-baToN (Tang et al, 2020), and the proximity labelling systems FAP-DAPA (Carpenter et al, 2020), LIPSTIC (Pasqual et al, 2018) and uLIPSTIC (Nakandakari-Higa et al, 2024). One limitation of these powerful systems is that they are based upon synthetic receptor-ligand interactions, so they require genetic modification of 'sender' cells and 'responding' cells (Morsut et al, 2016; Toda et al, 2020; Tang et al, 2020; Malaguti et al, 2022; Zhang et al, 2022).

Simpler systems engineer fluorescent molecules for both export by senders and uptake by surrounding cells. This is made possible by the fusion of signal peptides and cell-penetrating peptides, such as the HIV-1 transactivator of transcription (TATk), with fluorescent proteins. This concept has been exemplified by the PTD-GFP system for green-fluorescent protein (GFP) (Flinterman et al, 2009), and by the Cherry-niche (Ombrato et al, 2019, 2021) and its inducible version (CILP) (Zhang et al, 2023) for the red-fluorescent mCherry. The Cherry-niche system, in particular, has proven its efficacy in

[1]Centre for Regenerative Medicine, Institute for Regeneration and Repair, School of Biological Sciences, The University of Edinburgh, Edinburgh EH16 4UU, UK. [2]The Francis Crick Institute, London NW1 1AT, UK. [3]Centre for Discovery Brain Sciences, University of Edinburgh, Edinburgh EH8 9XD, UK. [4]Simons Initiative for the Developing Brain, The University of Edinburgh, Edinburgh EH8 9XD, UK. [5]Institute for Bioengineering, School of Engineering, University of Edinburgh, Edinburgh EH8 3DW, UK. [6]UK Centre for Mammalian Synthetic Biology, University of Edinburgh, Edinburgh EH9 3BF, UK. [7]Present address: Centre for Engineering Biology, Institute of Quantitative Biology, Biochemistry and Biotechnology, School of Biological Sciences, The University of Edinburgh, Edinburgh EH9 3FF, UK. ✉E-mail: sally.lowell@ed.ac.uk

characterising the tumour niche (Ombrato et al, 2019; Nolan et al, 2022). Nonetheless, access to alternative systems for the secretion and uptake of fluorescent proteins would enhance flexibility in the design of neighbour-labelling experiments.

Supercharged GFP has previously been used as a macromolecule-delivery system (McNaughton et al, 2009; Cronican et al, 2010; Thompson et al, 2012a). It is engineered to have a theoretical net charge of +36 (+36GFP) (Lawrence et al, 2007), facilitating interaction with cell membranes and subsequent endocytosis (McNaughton et al, 2009; Thompson et al, 2012b). +36GFP is efficiently taken-up by mammalian cells (McNaughton et al, 2009), and so proteins fused to +36GFP can be delivered into cells, outperforming fusions to commonly used cell-penetrating peptides (such as the TATk peptide and the Penetratin peptide) in several cell lines (Cronican et al, 2010).

In this study, we set out to examine whether +36GFP could be re-purposed for neighbour labelling. To maximise the range of applications, we explored strategies for signal amplification, with the aim of developing a sensitive system that could unambiguously identify even relatively transient interactions and be amenable to live imaging. We also sought simple options for changing the colour of labelling so that the system could be combined with any existing fluorescent reporters, and a customisable design of exchangeable modules that would maximise the flexibility of experimental design. We aimed to design a system that could be readily delivered to intact tissues or embryos using electroporation or viral infection, and which could be stably implemented in cultured cells, including pluripotent cells.

Neighbour-labelling tools open up opportunities for testing how differentiation in one cell influences differentiation in surrounding cells. Between day 4.5 and day 5.5 of mouse development, cells take the first step towards embryonic lineage commitment by exiting from naïve pluripotency (Nichols and Smith, 2009; Smith, 2017). It is not known whether the timing of this transition is coordinated between neighbouring cells, for example, to ensure that all cells complete the transition by E5.5. However, there is indirect evidence suggesting that this might be the case. For example, timely exit seems unlikely to depend only on cells sharing the same pro-differentiation environment: within initially homogenous monolayer cultures, some naïve cells exit the naïve state more than a day later than other cells within the same dish, despite being exposed to the same culture conditions, and independently of cell cycle status (Kalkan et al, 2017; Strawbridge et al, 2020; Jayaram et al, 2023). This observation also suggests that timely exit is not governed by a reliable intrinsic clock or by long-range synchronisation mechanisms.

We therefore hypothesised that short-range communication between cells helps to synchronise the rate of exit from naïve pluripotency, akin to a community effect ('local-synchronisation hypothesis'). We also considered two alternative possibilities: that local communication desynchronises the rate of exit ('local de-synchronisation hypothesis'), or that the rate of exit in one cell is not influenced by the rate of exit in neighbouring cells. Until now, it has been challenging to distinguish these hypotheses because the timing of exit from naïve pluripotency is functionally defined using clonal commitment assays to establish changes in potency (Kalkan et al, 2017), limiting the utility of imaging-based approaches. Fluorescent neighbour labelling overcomes this challenge, making it possible to compare the differentiation status of close neighbours to that of more distant cells using functional clonal assays of commitment status rather than relying on molecular markers.

Here we present a novel system, called PUFFFIN, for fluorescent neighbour labelling in culture or in vivo. We use PUFFFIN to obtain evidence that pluripotent cells adjust the pace of differentiation to synchronise with their neighbours during exit from naïve pluripotency.

# Results

## PUFFFIN can unambiguously label surrounding cells

We set out to design a system that could facilitate unambiguous labelling of neighbours after delivery of a single plasmid (Fig. 1A). The supercharged green-fluorescent protein +36GFP (Fig. 1B), is a positively-charged version of GFP (Lawrence et al, 2007) that can be taken up by mammalian cells when exogenously introduced to cultures (McNaughton et al, 2009; Cronican et al, 2010; Thompson et al, 2012b). We, therefore, decided to assess whether +36GFP could form a core component of a neighbour-labelling system. We added the human serum albumin signal peptide (Dugaiczyk et al, 1982), which is cleaved upon secretion, to create a secreted form of +36GFP which we refer to as s36GFP.

To maximise the sensitivity, speed, and specificity of neighbour labelling, we sought a strategy to boost fluorescence intensity. To achieve this, we fused s36GFP to the ultra-bright green-fluorescent mNeonGreen (mNG) (Shaner et al, 2013) as a signal amplifier (Fig. 1B). This was the first step towards generating a system we call 'positive ultra-bright fluorescent fusion for identifying neighbours' (PUFFFIN).

The PUFFFIN plasmid also encodes mCherry-3NLS, which remains inside the nucleus and serves to distinguish secretors from surrounding cells that have taken up s36GFP-mNG (Fig. 1A,C,D). The PUFFFIN expression cassette is driven by the strong ubiquitous CAG promoter (Niwa et al, 1991; Dou et al, 2021) and cHS4 insulators flank the expression cassette to guard against silencing (Chung et al, 1993) (Fig. 1C).

We delivered the PUFFFIN plasmid to HEK293 cells and generated stable monoclonal cell lines. The resulting "secretor" cell lines expressed mCherry-3NLS and s36GFP-mNG at levels that clearly distinguish secretors from unmodified cells by flow cytometry (Figs. 1E and Fig. EV1A).

To test the neighbour-labelling capacity of s36GFP-mNG, we mixed a minority of unmodified cells with an excess of secretors (1:4 ratio) to ensure that unmodified cells were likely to be in proximity to secretors. Within 48 h, over 95% of the unmodified cells became clearly s36GFP-mNG⁺ (Figs. 1E and Fig. EV1A–C). As expected, reducing the proportion of secretor cells reduced the extent of labelling in unmodified cells (Fig. EV1B). Secretors can also efficiently transfer s36GFP-mNG to UCH1N chordoma cells, mShef7 human embryonic stem (ES) cells, and E14tg2a mouse ES cells (Fig. EV1D).

This demonstrates that mammalian cells expressing the PUFFFIN plasmid unambiguously label surrounding cells.

## PUFFFIN labelling is confined to the local cell neighbourhood

A successful neighbour-labelling system depends on labelling being limited to nearby cells. To test this, we plated a small number of

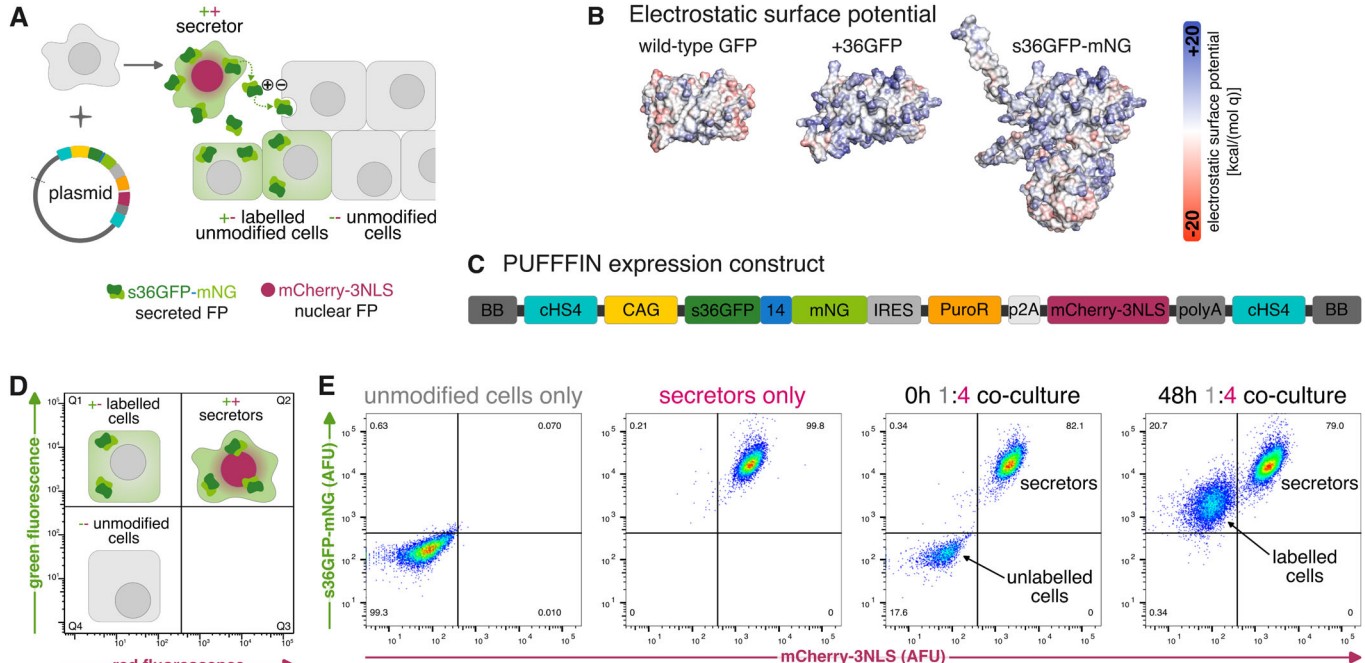

**Figure 1. Design and proof of concept of PUFFFIN labelling.**

(A) PUFFFIN cell lines can be generated by transfection with the random integration plasmid containing the PUFFFIN expression construct. The resulting cells stably express s36GFP-mNG which is transferred to nearby cells while retaining nuclear mCherry-3NLS. (B) Electrostatic surface potential is shown for wild-type GFP (PDB ID: 4kw4), +36GFP (AlphaFold predicted structure), and s36GFP-mNG (AlphaFold predicted structure). (C) The PUFFFIN expression cassette on the random integration plasmid contains the following parts: cHS4 insulators on both ends, the strong ubiquitous CAG promoter, the labelling protein s36GFP-mNG fused by a flexible 14 amino acid linker, an IRES for translation of downstream components, the puromycin resistance gene, a p2A self-cleaving peptide, nuclear mCherry-3NLS, and a synthetic polyA with a mammalian terminator. (D) Mock flow cytometry plot showing the three populations expected for PUFFFIN labelling. (E) Unmodified mCherry⁻ cells become s36GFP-mNG⁺ after mixing with mCherry⁺ s36GFP-mNG⁺ secretors in a 1:4 unmodified cell:secretor co-culture. 10,000 cells were analysed for the control panels and 30,000 cells for the 48 h 1:4 co-culture. Three independent experiments were performed, and a representative set is shown. +36GFP, green-fluorescent protein with a theoretical net charge of +36; 14, 14 AA linker; BB, backbone; CAG, CMV early enhancer/chicken β-actin promoter; cHS4, chicken hypersensitive site 4 insulator; FP, fluorescent protein; GFP, green-fluorescent protein; IRES, internal ribosome entry site; NLS, nuclear localisation signal; mNG, mNeonGreen; polyA, polyadenylation tail; PuroR, puromycin resistance gene; s36GFP, +36 green-fluorescent protein with an N-terminal secretion signal.

secretors among a large excess of unmodified cells (50:1 ratio) and performed time-lapse imaging.

Over time, green fluorescence became associated with unmodified cells that were in the proximity of secretors, while more distant cells remained unlabelled (Figs. 2A; Mov. EV1). This indicates that the s36GFP-mNG label is delivered only to nearby cells. To further assess the distance of labelling, we established a border between a secretor population and a population of unmodified cells, as illustrated in Fig. 2B. The two populations were plated in two different wells separated by a removable two-well insert and cultured to confluency, upon which the insert was removed so both populations could expand towards each other.

s36GFP-mNG labelling became visible in unmodified cells that had reached the immediate neighbourhood of the secretor population but was not visible further from the border between the two populations in Fig. 2C and Mov. EV2. Using the last time point of the live imaging, we quantified the red and green fluorescence intensity from the border starting with the last secretor (identified by nuclear mCherry). Whereas mCherry was not detected beyond the last secretor, the GFP intensity decreased gradually with distance, reaching its half maximum at 35 μm from the border (Fig. 2D). We determined the average cell diameter as 11.6 ± 1.6 μm and conclude that clear labelling is detectable up to

approximately three cells away from the border with weaker labelling extending a few cells beyond this (we note that this experiment is not designed to determine whether or not the green-fluorescent fusion protein can travel directly across several cell diameters, because the distance of labelling from the border will be dependent on cell division and cell migration as well as direct transfer of label).

Taken together, these data confirm that clearly detectable label is delivered only to cells that are in close proximity to secretors.

## PUFFFIN labelling is transferred rapidly between cells

During the time-lapse experiments described above (Fig. 2A,C) we could detect green fluorescence within neighbours of secretors within 30 min. To further assess how long it takes for the s36GFP-mNG label to become detectable, we mixed unmodified cells with secretors at different ratios and then used centrifugation to bring them together rapidly at high density. We incubated the pelleted co-culture for 15, 30, 45, or 60 min, and then analysed green fluorescence within the mCherry-negative unmodified cell fraction at various time points (Fig. 2E).

With an excess of PUFFFIN secretors (1:9 ratio), more than half of the unmodified cells (56%) were labelled with s36GFP-mNG

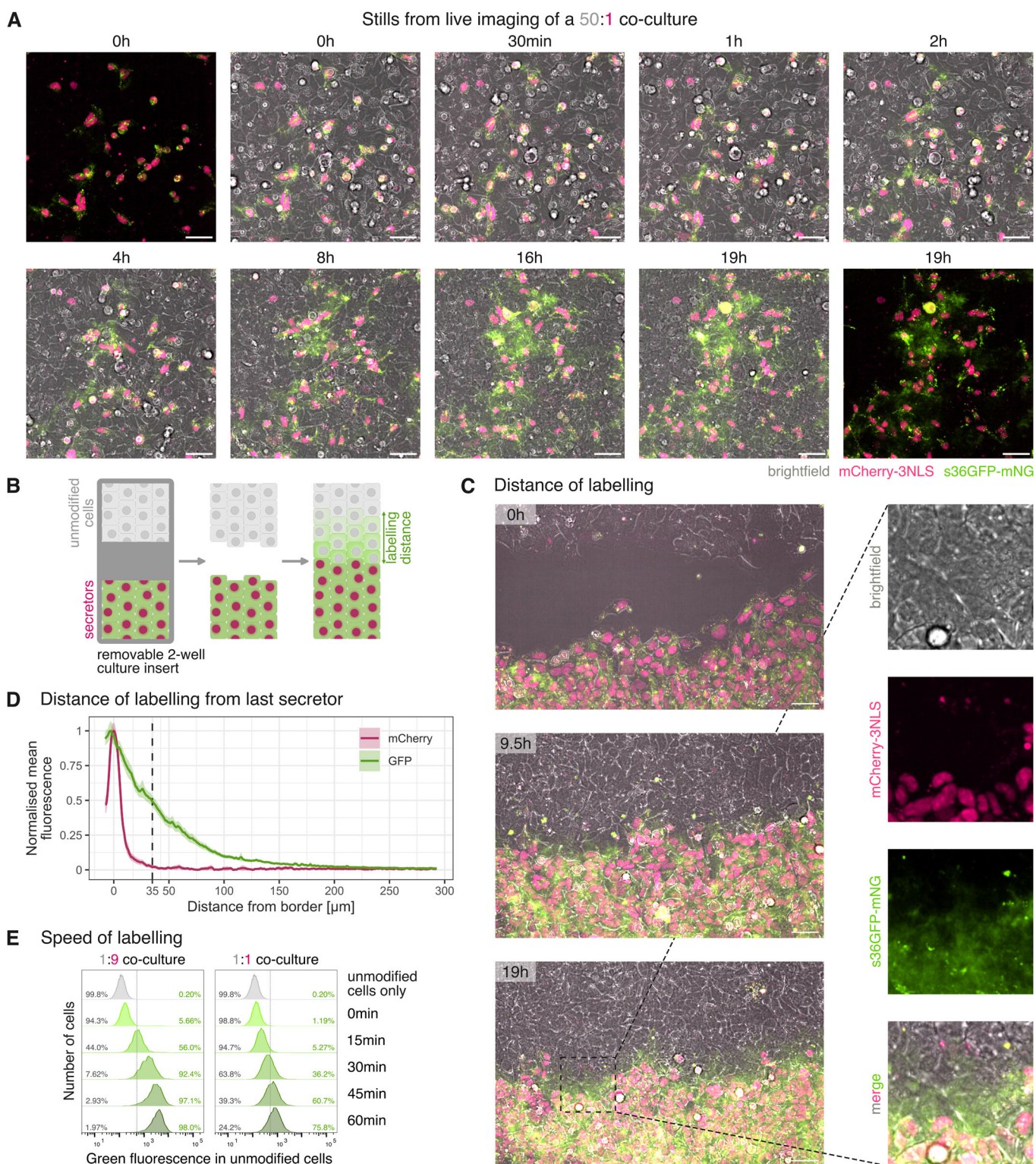

**A** Stills from live imaging of a 50:1 co-culture

0h    0h    30min    1h    2h

4h    8h    16h    19h    19h

brightfield    mCherry-3NLS    s36GFP-mNG

**B**

unmodified cells
labelling distance
secretors
removable 2-well culture insert

**C** Distance of labelling

0h

9.5h

19h

brightfield

mCherry-3NLS

s36GFP-mNG

merge

**D** Distance of labelling from last secretor

mCherry
GFP

Normalised mean fluorescence

Distance from border [μm]

**E** Speed of labelling

1:9 co-culture    1:1 co-culture

Number of cells

| | 1:9 co-culture | 1:1 co-culture | |
|---|---|---|---|
| 99.8% | 0.20% | 99.8% | 0.20% | unmodified cells only |
| 94.3% | 5.66% | 98.8% | 1.19% | 0min |
| 44.0% | 56.0% | 94.7% | 5.27% | 15min |
| 7.62% | 92.4% | 63.8% | 36.2% | 30min |
| 2.93% | 97.1% | 39.3% | 60.7% | 45min |
| 1.97% | 98.0% | 24.2% | 75.8% | 60min |

Green fluorescence in unmodified cells

within 15 min of mixing them with secretors, and nearly all of them (98%) were labelled within 1 h. For the 1:1 co-culture, labelling was slightly slower: only around 5% of cells were labelled after 15 min, increasing to over 75% after 1 h.

In co-cultures where secretor cells formed a small minority (50:1 unmodified cells:secretor cells), labelling could be detected at low

levels within the first hour. In this case, labelling was restricted to a minor fraction of cells, consistent with labelling being restricted to the close proximity of these sparse secretors (Fig. EV2A).

We also examined the speed of labelling in unmodified cells mixed with an excess of secretors (1:9 ratio) and then allowed to attach onto tissue culture plastic (Fig. EV2B). In this case, labelling

    

◀ **Figure 2. Distance and speed of PUFFFIN labelling.**

(A) Stills from live imaging of a 50:1 unmodified cell:secretor co-culture over 19 h. Scale bars are 50 µm. (B) Schematic of the experimental setup for testing the PUFFFIN labelling range. Secretors and acceptors are seeded separately and cultured to confluency in a removable cell culture insert. Insert is removed to allow both cell types to close the gap. Labelling at the border can be observed by fluorescence microscopy. (C) Stills from live imaging of a border experiment shows close labelling range. Scale bars are 50 µm. (D) Distance of labelling shown as quantification of mCherry and GFP intensity from live imaging data at 19 h time point. Fluorescence profiles with a width of 300 µm starting with the last secretor were measured in 10 µm intervals over an 1890-µm-long border between secretors and unmodified cells. Mean for 189 profiles and the 95% confidence interval of the mean for GFP (green) and mCherry (pink). (E) Speed of labelling is dependent on co-culture ratio and time: flow cytometry analysis of different unmodified cell:secretor ratios in a pelleted suspension co-culture as a time course experiment. Grey numbers (bottom right of each plot) are the percentage of unmodified cells in Q4 (see Fig. 1D), green numbers (bottom left of each plot) are the percentage of unmodified cells in Q1 (see Fig. 1D). 50,000 cells were analysed for each sample. Three independent experiments were performed, and a representative set is shown.

was detected within the majority of cells (71%) within 2 h of plating the cells.

We conclude that PUFFIN labelling can be transferred rapidly to neighbouring cells at levels that can be detected by flow cytometry or by live imaging and that labelling intensity is dependent on the co-culture ratio.

## The PUFFFIN system has a modular design to combine manipulation of cell function with customisable neighbour labelling within a single plasmid

To achieve ready customisability, we use the extensible mammalian modular assembly (EMMA) platform (Martella et al, 2017) to combine all components of PUFFFIN within a single plasmid comprised of modular exchangeable parts (Fig. 3A).

For example, to make a version of PUFFFIN that is compatible with cells that already contain red-fluorescent reporters, we exchanged the mCherry-3NLS transgenes to a cytoplasmic blue-fluorescent tagBFP (Subach et al, 2008) (Figs. 3B and EV3), and at the same time replaced the IRES with a t2A self-cleaving peptide (Liu et al, 2017). We used this customised construct to generate a stable polyclonal line of secretors in which the vast majority of cells (>95%) are s36GFP-mNeonGreen⁺ tagBFP⁺ (Fig. EV3), without the need for selecting clonal cell lines.

To expand the utility of the system for functional experiments, the puromycin resistance gene can be exchanged for any transgene of interest for manipulating or monitoring cell behaviour (Fig. 3C) (the selection gene is dispensable because stably expressing secretors can be purified based on fluorescent signal rather than antibiotic selection).

Similarly, the CAG promoter can be switched to an inducible or a cell type-specific promoter for regulatable or tissue-specific expression of the PUFFFIN construct (Fig. 3D). This offers the opportunity to manipulate cell behaviour in a specific set of cells while simultaneously delivering a fluorescent label to their neighbours.

## The PUFFFIN system can be customised for colour-of-choice labelling using HaloTag® technology

We demonstrate above (Figs. 1 and 2) that mNeonGreen is an effective signal amplifier that facilitates bright neighbour labelling using PUFFFIN. However, s36GFP-mNeonGreen labelling is not compatible with cells that already express GFP, such as green-fluorescent reporter lines. We, therefore, sought an approach to further customise the PUFFFIN construct so that labelling can be

performed in any colour of choice without the need for additional genetic manipulation of the PUFFFIN plasmid.

To achieve this, we replaced the coding sequence of mNeon-Green with a HaloTag® (Los et al, 2008), illustrated in (Figs. 4A,B and EV4A), creating an s36GFP-HaloTag fusion protein ('PUFF-Halo') that can bind any HaloTag fluorescent ligand (England et al, 2015; Cook et al, 2023) (Fig. 4C). We used this new construct to generate stable PUFFHalo secretor lines.

We first confirmed that the residual green fluorescence from the s36GFP was relatively dim in the absence of any amplifier, and therefore likely to be compatible with GFP-based reporters, and unlikely to override our ability to customise the colour of PUFFFIN labelling using HaloTag-binding dyes (Fig. 4D). We then tested whether we could customise the colour of PUFFFIN labelling using fluorescent HaloTag ligands. We cultured secretors in the presence or absence of far-red (JF646) or green (OregonGreen) fluorescent dyes and observed a strong shift in fluorescence following dye addition (Figs. 4E and EV4B). This suggests that PUFFHalo can be labelled with different fluorophores and detected by flow cytometry.

We next tested the ability of PUFFHalo to label neighbours, by mixing secretors with unmodified cells in a 1:1 co-culture at high density for 48 h. More than 99% or 85% of unmodified cells were labelled by the PUFFHalo in the presence of far-red or green HaloTag fluorescent ligands, respectively (Figs. 4F and EV4C). This suggests PUFFHalo can be transferred to unmodified neighbours of secretor cells, and that labelled neighbours can be clearly identified by flow cytometry.

Finally, we investigated the potential of PUFFHalo for live imaging applications, by mixing a low number of secretors with an excess of unmodified cells (50:1 ratio) for 48 h and imaging the cells following the addition of HaloTag dyes. Labelling was clearly visible within neighbours of mCherry⁺ secretors and could also be seen decorating the cell membrane and underlying substrate (Figs. 4G and EV4D,E). This suggests that PUFFHalo can be used to monitor neighbour labelling in live cells in real time.

We conclude that PUFFHalo is readily customisable using HaloTag-compatible dyes to offer labelling in a colour of choice, and that the Halo-compatible dyes can further enhance the sensitivity of neighbour labelling.

## The PUFFFIN system can be used for neighbour labelling in vivo

We next asked if PUFFFIN could be delivered to tissues in vivo or ex vivo. To facilitate packaging into a viral vector or transduction by electroporation, we reduced the size of the PUFFFIN construct

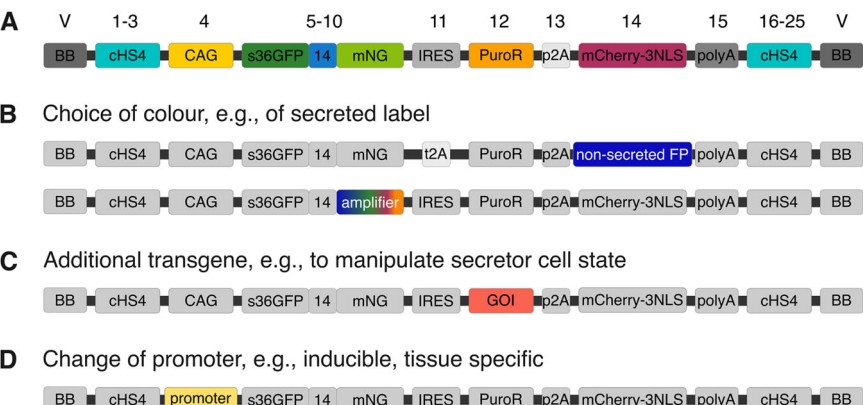

**Figure 3. Modular design of PUFFFIN plasmid assembly.**

(A) Standard PUFFFIN expression construct. The numbers above the plasmid chart stand for positions in the EMMA platform. (B) Assembly of different parts allows, for example, a combination of the s36GFP-mNG label with a non-secreted fluorescent protein (FP) in another colour. The PUFFFIN label could be a fusion of s36GFP with an amplifier other than mNeonGreen to change the colour of the label. (C) The selection gene could be replaced by a gene of interest, (D) and the ubiquitous CAG promoter by a cell type-specific promoter.

by removing the selection cassette and insulator sequences to generate minimal constructs (Fig. 5A).

We packaged the smallest minimal PUFFFIN construct into a lentivirus (Fig. EV5A) and used it to transduce mouse brain organotypic slices (Fig. EV5B) (Eigel et al, 2019) at low titre ($8.9 \times 10^6$ TU/mL), aiming to obtain mosaic and sparse expression. Transduced brain slices were cultured for a further 10 days, then examined for expression of s36GFP-mNeonGreen and mCherry-3NLS. We found that strongly mCherry-positive nuclei were surrounded by s36GFP-mNeonGreen signal (Fig. EV5C), indicating successful lentiviral delivery of the PUFFFIN system, although the complex morphology of neural cells made it difficult to assess whether the label had spread to neighbours in this experimental context.

We then delivered the minimal PUFFFIN s36GFP-HaloTag (minimal PUFFHalo) construct by electroporation into Hamburger Hamilton (HH) stage 10 chick embryos. We targeted the caudal ectoderm (Fig. 5B) using a low concentration of plasmid, aiming to obtain mosaic and sparse expression. Embryos were allowed to develop for a further 24 h, then examined for expression of s36GFP-HaloTag and mCherry-3NLS at HH15. Similar to the results obtained with brain slices (Fig. EV5C), strongly mCherry-positive cells within the chick caudal ectoderm were surrounded by s36GFP-HaloTag label (Fig. 5C). Label appeared to have spread into mCherry-negative cells close to transduced cells, but not into more distant cells (Fig. 5C). Distance of labelling was restricted to around 3–4 cell diameters, similar to the distance of labelling previously observed in cell cultures (Fig. 5C: compare with Figs. 2 and 4).

To confirm the transfer of the label into neighbours, we performed flow cytometry analysis on dissociated trunk cells 24 h after electroporation. Transduced cells were identified by expression of nuclear mCherry, with the strongest mCherry signal being associated with the highest levels of s36GFP-HaloTag, as expected (Fig. 5D). s36GFP-HaloTag could also be detected in a small population of mCherry-negative cells, at levels comparable to

labelling observed in cell culture experiments where secretors were in a small minority (Fig. 5D: compare with Fig. EV2A).

We conclude that the PUFFFIN system can be delivered by lentiviral infection or electroporation and can be used for neighbour labelling in vivo.

## Implementation of PUFFFIN in pluripotent cells

Pluripotent cells provide a simple and tractable experimental system for studying cell fate decisions, so we asked whether PUFFFIN could be implemented in mouse ES cells (mESCs). We delivered the PUFFHalo plasmid to mESCs and generated stable monoclonal cell lines. The resulting secretor cell lines expressed mCherry-3NLS and s36GFP-HaloTag at levels that clearly distinguished secretors from unmodified cells by flow cytometry (Fig. 5E).

To test the neighbour-labelling capacity of s36GFP-HaloTag in pluripotent cells, we mixed unmodified cells with secretors in various ratios. When secretors were in the minority (9:1 ratio), a minor subpopulation of unmodified cells became labelled, whereas when secretors formed the majority (1:9 ratio), almost all unmodified cells became labelled (Fig. 5E). In 9:1 co-cultures, s36GFP-HaloTag label appeared to have spread into mCherry-negative cells close to secretors, but not into more distant cells (Fig. 5F), in keeping with observations made in HEK293 cells (Figs. 2 and 4) and chick embryos (Fig. 5C).

We conclude that PUFFFIN can be used for neighbour labelling in pluripotent cells, opening up opportunities for studying how cells influence their neighbours during differentiation.

## Pluripotent cells adjust the pace of differentiation to coordinate with their neighbours during exit from naïve pluripotency

Cells exit naïve pluripotency at different rates, even within the same culture (Kalkan et al, 2017). It remains an open question whether

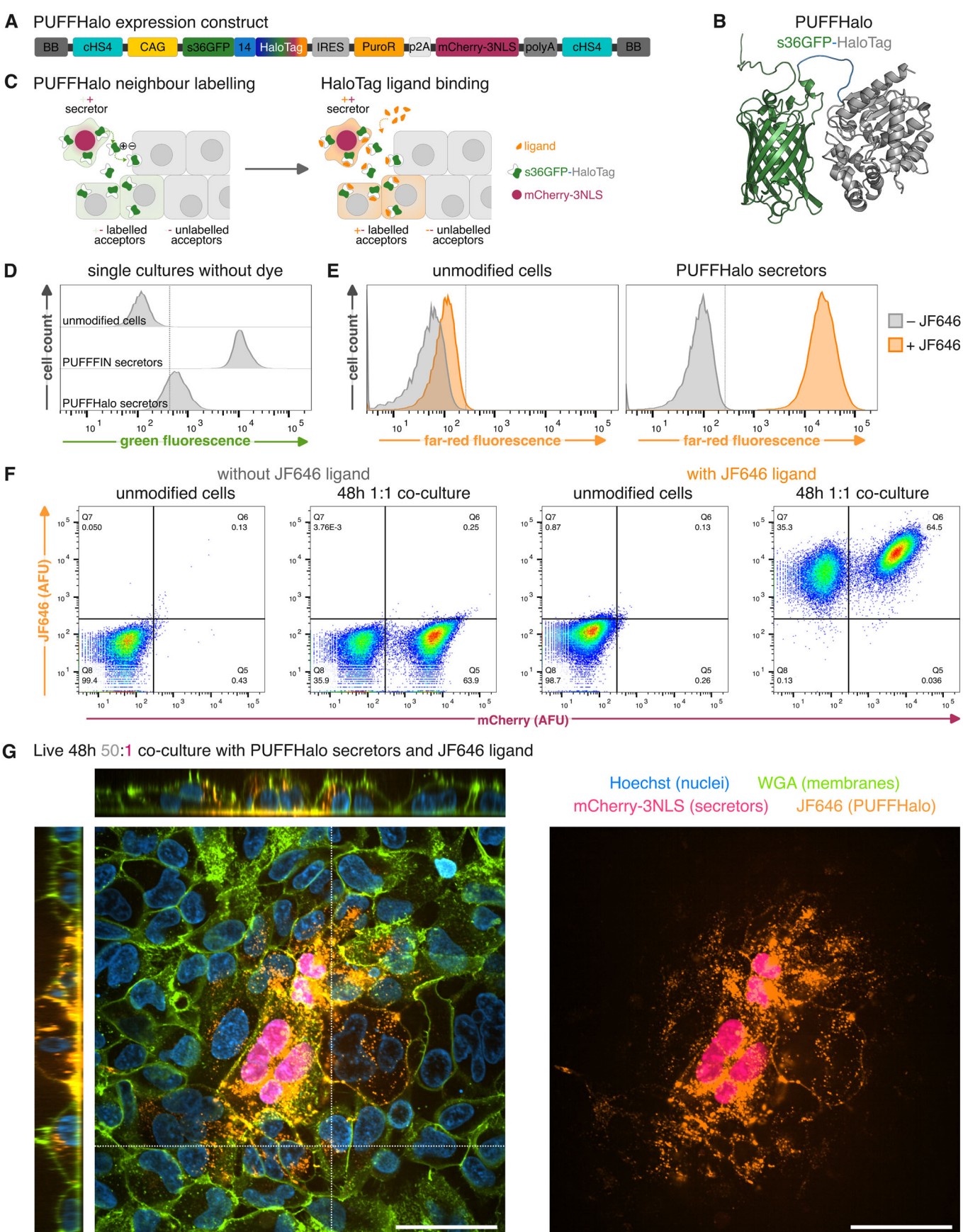

**A** PUFFHalo expression construct

**B** PUFFHalo s36GFP-HaloTag

**C** PUFFHalo neighbour labelling — HaloTag ligand binding

**D** single cultures without dye

**E** unmodified cells — PUFFHalo secretors

**F** without JF646 ligand — with JF646 ligand

**G** Live 48h 50:1 co-culture with PUFFHalo secretors and JF646 ligand

Hoechst (nuclei)  WGA (membranes)
mCherry-3NLS (secretors)  JF646 (PUFFHalo)

◄ **Figure 4. Combination of PUFFFIN with HaloTag technology enables colour-of-choice neighbour labelling.**

(A) The modularity of PUFFFIN allows an assembly termed "PUFFHalo" that contains HaloTag as an amplifier instead of mNeonGreen. (B) As an alternative to the s36GFP-mNG label, s36GFP (green) was fused to HaloTag (grey) as a signal amplifier by a 14 amino acid-long linker (blue) (AlphaFold predicted structure). (C) Neighbour labelling is possible with PUFFHalo but has only low green fluorescence on its own. The label can be bound by HaloTag ligands, e.g. dyes in different colours, which amplifies the labelling signal. (D) Comparison of unmodified cells, s36GFP-mNG secretors and PUFFHalo secretors shows low green-fluorescence of the unstained PUFFHalo. (E) Unmodified cells and PUFFHalo secretors were incubated for 2 h with or without the HaloTag ligand Janelia Fluor (JF646). (F) Flow cytometry of 48 h 1:1 co-cultures of unmodified cells and PUFFHalo secretors were incubated for 2 h with or without the HaloTag ligand JF646. For (D–F), three independent experiments were performed, and a representative set is shown. (G) Live imaging of a 48 h 50:1 co-culture of unmodified cells and PUFFHalo secretors with the HaloTag ligand JF646, with Hoechst nuclear staining and WGA488 membrane staining. Dotted lines mark regions selected for orthogonal views. The left image shows the merge of all channels, right only far-red JF646 (PUFFHalo) and red (mCherry-3NLS, secretor nuclei). Scale bars are 50 µm, image depth is 11.5 µm.

cells coordinate the pace of differentiation with nearby cells. For example, do the faster-differentiating cells signal to nearby cells (but not more distant cells) to 'speed up', in order to limit the degree of asynchrony (local-synchronisation hypothesis)? Or does the opposite happen, with faster-differentiating cells signalling to nearby cells to 'slow down', in order to diversify the rate of exit (local de-synchronisation hypothesis)? The third possibility is that the rate of differentiation is not locally coordinated in either direction, i.e. cells pay no attention to the rate of differentiation of their neighbours.

To distinguish between these three hypotheses, we made use of a well-established experimental system (Betschinger et al, 2013; Kalkan et al, 2017; Mulas et al, 2023) in which naïve cells are released from naïve pluripotency culture conditions (2iLIF culture medium) and given the opportunity to differentiate over the course of 48 h in N2B27 differentiation medium. At any given time during this transition, clonal assays can be used to measure the proportion of cells that have irreversibly committed to exit the naïve pluripotent state. These assays are performed by plating cells at clonal density in 2iLIF: naïve cells are able to form colonies under these conditions, while cells that have committed to exit the naïve state are not. Using this approach, it has been shown that cells exit naïve pluripotency asynchronously over the course of 48 h (Kalkan et al, 2017). Based on this experimental setup, we designed an experiment using PUFFFIN neighbour labelling to ask whether cells coordinate the rate of differentiation with their neighbours (Fig. 6A).

To represent 'faster-differentiating cells', we released s36GFP-HaloTag secretor mESCs into N2B27 for 24 or 48 h, to initiate exit from naïve pluripotency. We seeded these pre-differentiated secretor cells among an excess of naïve unmodified cells (9:1 ratio) to create a situation where secretors were at a more advanced stage of differentiation than their unmodified neighbours. As a control, we seeded naïve secretors among an excess of naïve unmodified cells (9:1 ratio), to create a situation where secretors were at the same stage of differentiation as their unmodified neighbours. All co-cultures were then cultured for 24 h in N2B27 (Fig. 6A). To establish the baseline colony-forming efficiency (CFE) for cells not exposed to differentiation conditions, we co-cultured secretors with unmodified cells (9:1 ratio) in 2iLIF throughout.

We confirmed that PUFFFIN components remain expressed, and that labelling continues to operate during differentiation, with labelling restricted to neighbours of mCherry-positive cells as expected (Fig. 6B). Flow cytometry could distinguish secretors (mCherry, s36GFP-HaloTag double positive), neighbours (s36GFP-HaloTag single positive), and non-neighbours (double negative) (Fig. EV6A).

We sorted secretors, neighbours, and non-neighbours from all co-cultures and plated them at clonal density in 2iLIF (250 live cells per dish). We first examined data from control co-cultures in which secretors had not been pre-differentiated. If PUFFFIN labelling itself has adverse effects on colony-forming efficiency then we would expect to see significantly fewer colonies arising from labelled cells than unlabelled cells, but this was not the case (labelled cells formed $90.1 \pm 12.4$ colonies, unlabelled cells formed $93.8 \pm 20.5$ colonies) (Fig. EV6B).

We next examined data from co-cultures in which secretors had been pre-differentiated in N2B27 for 24 or 48 h then further differentiated for an additional 24 h as co-cultures (experimental design illustrated in Fig. 6A,C). We first focused on the secretors taken from these co-cultures. As expected, almost no colonies formed from these cells ($2.9 \pm 1.4$ colonies formed from secretors pre-differentiated for 24 h then differentiated in co-culture for a further 24 h; $1.4 \pm 0.4$ colonies formed from secretors pre-differentiated for 48 h then differentiated in co-culture for a further 24 h). This is consistent with previous reports that almost all cells exit naïve pluripotency within 48 h of culture in N2B27 (Kalkan et al, 2017).

We next focused on the unmodified (non-secretor) cells within the N2B27 co-cultures: in all cases, these had been exposed to N2B27 for only 24 h, during the co-culture period (Fig. 6A). It has previously been reported that around 2/3 of cells can form colonies in 2iLIF following 25 h culture in N2B27, when compared with baseline CFE of cells never exposed to differentiation conditions (Kalkan et al, 2017). We first examined data from cells distant from pre-differentiated cells (non-neighbours/unlabelled cells): around 60% of these non-neighbours form colonies (61% from 24h-pre-differentiated co-cultures and 58% from 48h-pre-differentiated co-cultures) when compared to baseline CFE (Fig. EV6B). We then examined neighbours of secretors (labelled cells) in control N2B27 co-cultures: 63% formed colonies when compared to baseline CFE (Fig. EV6B). Therefore, in all these cases, cells that had <u>not</u> been in close proximity to pre-differentiated cells exit naive pluripotency at a rate comparable with previous reports (Kalkan et al, 2017).

In contrast, neighbours of pre-differentiated cells tended to form fewer colonies: 46% for neighbours of 24h-pre-differentiated cells, or 41% for neighbours of 48h-pre-differentiated cells, when compared to baseline CFE (Fig. EV6B). This suggests that neighbours of pre-differentiated cells may tend to commit to exit naïve pluripotency at a faster-than-normal rate.

To more closely examine differences in the behaviours of neighbours vs non-neighbours, we plotted the ratio of colony-forming efficiencies (CFE) for neighbours vs non-neighbours sorted from the same dish (N:NN ratio). For control co-cultures, the average N:NN ratio was close to 1 (0.96) and was not significantly

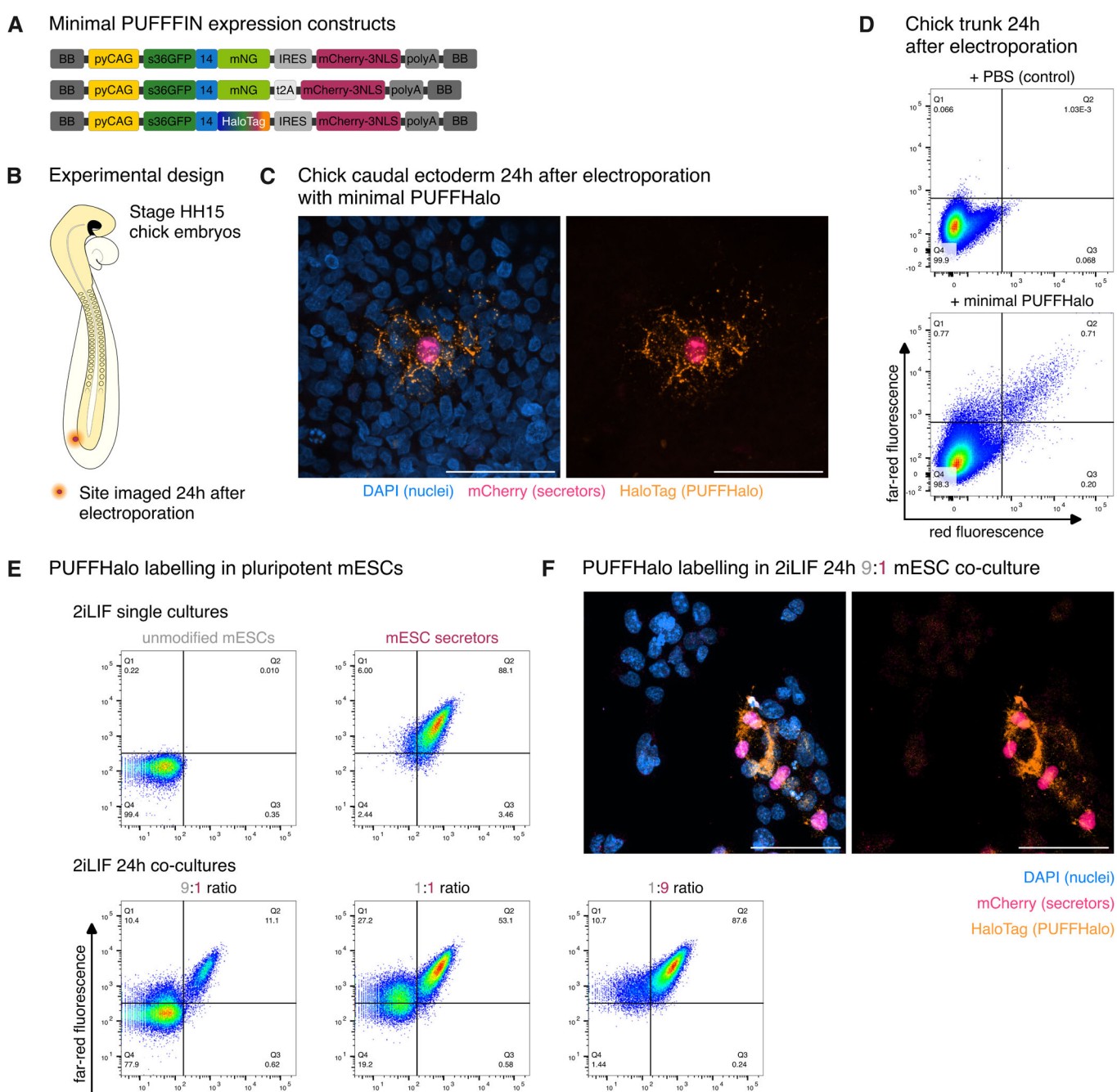

**Figure 5. PUFFFIN labelling in model systems of development.**

(A) Three versions of the minimal PUFFFIN expression construct, including a minimal PUFFHalo, were designed for applications where plasmid size is a limiting factor for efficient delivery, e.g. in vivo electroporation. (B) Schematic showing a Hamburger Hamilton (HH) stage 15 embryo indicating the site imaged 24 h after delivery of the minimal PUFFHalo plasmid. (C) Confocal imaging of trunk ectoderm of an HH15 chicken embryo, electroporated in the caudal ectoderm at HH10 with minimal PUFFHalo and incubated ex ovo for 24 h. Embryos were incubated for 1 h with the HaloTag ligand JF646 (orange) before staining with DAPI (cyan) and mCherry nanobody to boost the fluorescence signal (pink). The left image shows the maximum intensity projection of all channels, right only far-red JF646 (PUFFHalo) and red (mCherry-3NLS, secretor nuclei). Scale bars are 50 μm, image depth is 13.5 μm. (D) Flow cytometry of live cells from pooled dissociated chick trunks (3–5 trunks per sample) for control embryos (electroporated with PBS) and for embryos electroporated with minimal PUFFHalo. Two independent experiments were performed, and a representative set is shown. (E) Flow cytometry of mESCs in 2iLIF as single cultures or 24 h co-cultures of unmodified cells and PUFFHalo secretors (different ratios as indicated in Figure), incubated for 2 h with the HaloTag ligand JF646 (far-red). Three independent experiments were performed, and a representative set is shown. (F) Immunofluorescence microscopy of a 24 h 9:1 mESC co-culture in naïve pluripotency medium (2iLIF) shows PUFFHalo labelling around secretors. Scale bars are 50 μm.

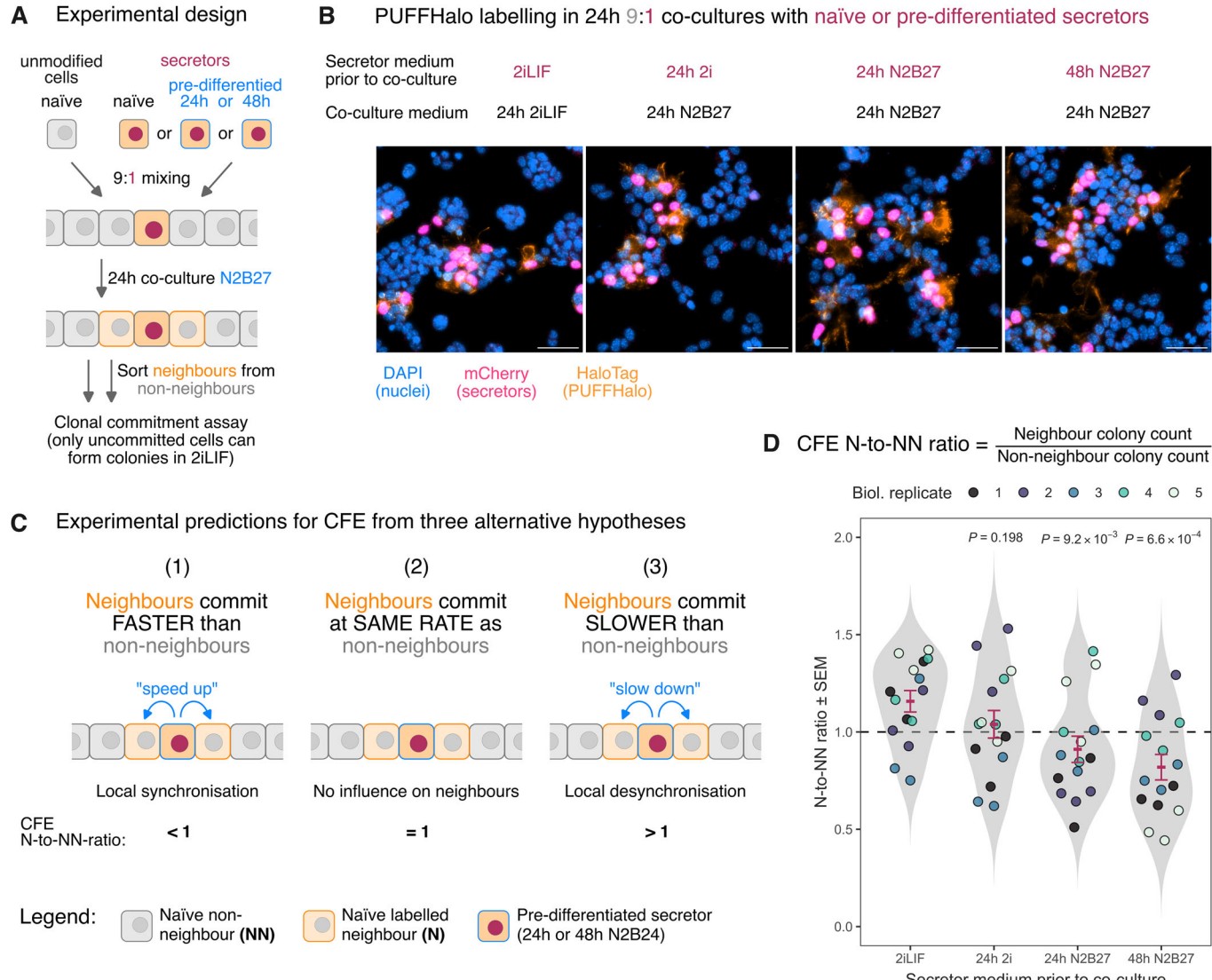

**Figure 6. Pluripotent cells adjust the pace of differentiation to coordinate with their neighbours during exit from naïve pluripotency.**

(A) Experimental design for co-culturing naïve unmodified cells with a minority of either naïve or pre-differentiated PUFFHalo secretors for 24 h. Secretors, their neighbours and non-neighbours can then be isolated by FACS and screened in functional assays, e.g. testing for pluripotency by clonal commitment assay where only uncommitted cells can form colonies. (B) In all four experimental conditions, PUFFHalo labelling is effective in a 24 h 9:1 co-culture as shown by IF microscopy. Scale bars are 50 μm. (C) Three alternative hypotheses can be considered for the experimental outcome of testing the colony formation efficiency (CFE) of neighbours and non-neighbours of naïve or pre-differentiated secretors in the clonal commitment assay. (D) CFE Neighbour-to-Non-neighbour ratio coloured by biological replicate. Red crossbars show the mean and standard error of the mean (SEM). *P* values represent Bonferroni-adjusted, two-sided Fisher's exact tests relative to the 2iLIF medium, using the sum of the colony counts averaged over five independent biological replicates. Five independent experiments with three non-independent replicates were performed.

different from baseline, indicating no consistent bias in CFE of neighbours vs. non-neighbours, as expected (Fig. 6D). In contrast, for cultures containing pre-differentiated secretors, the average N:NN ratio was significantly different from the baseline ratio (Fig. 6D). This indicates that pre-differentiated secretors influence the behaviour of their neighbours, and that this is unlikely to be attributable to artefactual effects of the labelling process. The ratio of colonies forming from neighbours:non-neighbours fell below, rather than above, 1 for co-cultures containing pre-differentiated secretors (Fig. 6D) in keeping with the predictions of the local-synchronisation hypothesis, and contrary to the predictions of the alternative hypotheses (Fig. 6C).

The observed difference in colony-forming efficiency between neighbours vs non-neighbours of pre-differentiated cells is unlikely to be due to differences in viability because only viable cells were sorted from co-cultures, and viability was also subsequently confirmed following sorting. The difference is also unlikely to relate to any adverse effect of the labelling process itself because there is no decrease in colony-forming efficiency in labelled neighbours taken from control cultures that lack pre-differentiated cells (Fig. 6D). We, therefore, conclude that our findings are most likely explained by neighbours of pre-differentiated cells being encouraged to exit naïve pluripotency by their pre-differentiated neighbours.

These experiments highlight particular strengths of PUFFFIN neighbour labelling. Firstly, it enables functional assays to be performed on live cells to compare properties of neighbours and non-neighbours, for example the clonal commitment assay that we use here. This facilitates sensitive detection of changes in cell potency that can occur prior to overt changes in the expression of molecular markers of differentiation, revealing changes that might not be detected by image analysis or other marker-based analysis. Secondly, the ability to directly compare neighbours and non-neighbours within the same dish makes it possible to sensitively detect significant changes that might not so easily be detected by classical bulk co-culture experiments.

We conclude that PUFFFIN neighbour labelling provides a sensitive, reliable, and flexible approach for detecting how cells influence their neighbours.

## Discussion

We demonstrate that positive ultra-bright fluorescent fusion for identifying neighbours (PUFFFIN) can be used to unambiguously, sensitively, and rapidly label cell neighbourhoods. We use PUFFFIN to obtain evidence that pluripotent cells coordinate the rate of differentiation with their neighbours.

The integration of PUFFFIN with HaloTag technology, PUFFHalo, enables colour-of-choice labelling with high signal-to-noise ratio without the need to reengineer the PUFFHalo plasmid. The HaloTag could, in principle, also be used for supporting extended live imaging (Cook et al, 2023), pulse-chase experiments (Yim et al, 2022), and specific degradation of the label using HaloPROTACS (Buckley et al, 2015). Our modular design can incorporate any transgene of interest to study non-cell-autonomous effects of experimentally induced changes in the identity of behaviour.

PUFFFIN offers a high level of customisability, a convenient single-plasmid delivery system, and receptor-independent transfer that does not require any modifications to the cells that receive the label. For example, the system could be delivered to embryos via electroporation in order to uncover non-cell autonomous responses to developmental regulators, or by viral transduction into tissues to uncover interactions between different tissues, or by grafting of PUFFFIN-engineered cancer cells into mice to characterise the tumour niche. Combining PUFFFIN with organoid technologies could generate powerful models for understanding how cell communication orchestrates self-organisation and regulates disease states.

We have not yet tested our system with tissue-specific drivers of PUFFFIN and it remains to be established how effective labelling will be at lower levels of expression. When delivering the PUFFFIN system by electroporation into chick embryos (Fig. 5) or lentiviral delivery into mouse brain organotypic slice cultures (Fig. EV5) we noticed that cells with low expression of mCherry sometimes lacked detectable PUFFFIN label, suggesting that relatively high levels of expression from the PUFFFIN construct may be necessary for effective labelling. The minimum level of PUFFFIN expression required for effective labelling is likely to depend on cell type and context, and, therefore, should be assessed for each experimental setup.

Another lesson learned from our experiments in brain slice cultures was that secretor cells with complex morphologies may be challenging to distinguish from their neighbours using imaging-based approaches alone. However, the modular design of the PUFFFIN construct (Fig. 3) should make it straightforward to overcome this problem by switching nuclear mCherry for a cytoplasmic or cell membrane-associated fluorescent protein that more clearly demarcates the outer limits of the secretor cells. Furthermore, we anticipate that any difficulties distinguishing secretors from labelled neighbours will be limited to imaging-based analyses and are unlikely to present any problems during flow cytometry.

A particular strength of fluorescent neighbour-labelling systems is that they allow for functional assays of live cells in experiments to compare the properties of neighbours vs non-neighbours. For example, we have used sensitive clonal assays of cell potency to address the question of whether mouse pluripotent cells coordinate the rate of differentiation with their neighbours during exit from naïve pluripotency. It is clear that diversity in potency emerges during exit from naïve pluripotency both in culture (Betschinger et al, 2013; Leeb et al, 2014; Kalkan et al, 2017) and in vivo (Acampora et al, 2017; Malaguti et al, 2019; Neagu et al, 2020): our findings suggest that cells may communicate with each other to limit the extent of this asynchrony. A number of elegant approaches have provided evidence for the synchronisation of differentiation between sibling pluripotent cells (Deathridge et al, 2019; Strawbridge et al, 2020; Chaigne et al, 2020); our findings indicate that there is also some degree of synchronisation between unrelated neighbours, suggesting that synchronisation may depend on cell-cell communication rather than only on clonal relationships.

PUFFFIN presents an effective and flexible approach for illuminating cellular neighbourhoods, offering many opportunities to discover how cells influence the functional properties of surrounding cells during development, homoeostasis, regeneration, and disease.

## Methods

### Reagents and tools table

| Reagent/Resource | Reference or source | Identifier or catalogue number |
|---|---|---|
| **Experimental models** | | |
| Human embryonic kidney (HEK) 293 | Blin lab | |
| E14Ju09 (129/Ola) mouse embryonic stem cells (subclone of E14tgta tested for high germline competence) | (Doetschman et al, 1987) | |
| mShef7 human embryonic stem cells | Blin lab | |
| | (Gouti et al, 2014) | |
| UCH1N human chordoma-derived cell line | Wilson lab | |
| | (Fujita et al, 2016) | |
| C57BL/6J (mouse) | Obtained from Charles River Laboratories | |

| Reagent/Resource | Reference or source | Identifier or catalogue number |
|---|---|---|
| Gallus gallus (chicken) | Obtained from Henry Stewart & Co. | |
| NEB® 5-alpha Competent *E. coli* (Subcloning efficiency) | New England Biolabs | #C2988J |
| NEB® 10-beta Competent *E. coli* (High efficiency) | New England Biolabs | #C3019I |
| PUFFFIN lentivirus | This study | |
| **Recombinant DNA** | | |
| YCe3736_HC_Amp_ccdB_receiver vector | Pollard lab, available on AddGene | #100637 |
| p_cHS4-CAG-s36GFP-mNG-IRES-PuroR-p2A-mCherry-3NLS-polyA-cHS4 | This study | pTL5 |
| p_cHS4-CAG-s36GFP-mNG-t2A-PuroR-p2A-BFP-polyA-cHS4 | This study | pTL50 |
| p_cHS4-CAG-s36GFP-HaloTag-IRES-PuroR-p2A-mCherry-3NLS-polyA-cHS4 | This study | pTL51 |
| p_pyCAG-s36GFP-HaloTag-IRES-mCherry-3NLS-polyA | This study | pTL52 |
| p_pyCAG-s36GFP-mNG-IRES-mCherry-3NLS-polyA | This study | pTL53 |
| p_pyCAG-s36GFP-mNG-t2A-mCherry-3NLS-polyA | This study | pTL54 |
| pLenti6_cppt-CMV-Puffling-opre | This study | pLenti6 |
| **Antibodies** | | |
| Mouse monoclonal anti-mNeonGreen (32F6, 1:500) | Proteintech | #17882377, |
| | | RRID: AB_2827566 |
| Rabbit polyclonal anti-HaloTag (1:500) | Promega | #G9281 |
| | | RRID: AB_713650 |
| Rabbit polyclonal anti-RFP (1:1000) | Rockland | #600-401-379, |
| | | RRID: AB_2209751 |
| Rat monoclonal anti-mCherry (1:1000) | Invitrogen | #M11217, |
| | | RRID: AB_2536611 |
| RFP-Booster Alexa Fluor® 568 (1:500) | ChromoTek | #rb2AF568 |
| | | RRID: AB_2827576 |
| **Chemicals, enzymes, and other reagents** | | |
| 2-mercaptoethanol | Gibco | #31350010 |
| 2-well removable culture-inserts | Ibidi | #80209 |
| 3-well removable culture-inserts | Ibidi | #80369 |
| 4-well μ-slides | Ibidi | #80426 |
| 8-well μ-slides | Ibidi | #80806-90 |
| Accutase | Invitrogen | #00-4555-56 |
| Amphotericin B | Gibco | #15290026 |

| Reagent/Resource | Reference or source | Identifier or catalogue number |
|---|---|---|
| B-27 Supplement | Gibco | #17504044 |
| Bovine serum albumin | Sigma-Aldrich | #A7906 |
| CHIR99021 | Axon Mechem | #1386 |
| D-glucose | Sigma-Aldrich | #G8769 |
| DAPI | Biotium | #40043 |
| DMEM/F-12 | Gibco | #21331020 |
| Donkey serum | Sigma-Aldrich | #D9663 |
| DRAQ7 | Abcam | #ab109202 |
| Dulbecco's Modified Eagle Medium (DMEM, high glucose, pyruvate) | Gibco | #41966029 |
| Dulbecco's phosphate-buffered saline (PBS) | Sigma-Aldrich | #D8537 |
| Earle's balanced salt solution (calcium, magnesium, phenol red) | Gibco | #24010043 |
| FACSmax cell dissociation solution | amsbio | AMS.T200100 |
| Fibronectin | Sigma-Aldrich | #F1141 |
| Fluoromount-G® mounting medium | SouthernBiotech | #0100-01 |
| Foetal calf serum (FCS) | Biosera | #FB-1280/500 |
| Formaldehyde 37–41% | Fisher Scientific | #F/1501/PB08 |
| Gelatine | Sigma-Aldrich | #G1890 |
| Glasgow minimum essential medium (GMEM) | Gibco | #1171035 |
| Glutamax™ supplement | Thermo Scientific | #35050-038 |
| Heat-inactivated horse serum, New Zealand origin | Gibco | #26050088 |
| Hibernate™-A medium | Gibco | #A1247501 |
| Hoechst 33342 | Apexbio | #A3472-APE |
| Hoechst 33342 solution | Thermo Scientific | #62249 |
| Janelia Fluor 646 (JF646) ligand | Promega | #GA1120 |
| L-glutamine | Invitrogen | #25030-024 |
| Laminin | Sigma-Aldrich | #L2020 |
| Leukaemia inhibitory factor (LIF) | produced in-house | |
| Lipofectamine 3000 transfection reagent | Invitrogen | #L3000001 |
| Millicell cell culture inserts (30 mm, hydrophilic PTFE, 0.4 μm) | Merck-Millipore | #PICM0RG50 |
| Minimum essential medium (MEM) | Gibco | #12360038 |
| Monarch plasmid DNA Miniprep Kit | New England Biolabs | #T1010 |
| N-2 supplement | Gibco | #17502048 |
| NEBridge Golden Gate Assembly Kit (BsmBI-v2) | New England Biolabs | #E1602L |
| Neurobasal medium | Gibco | #21103049 |
| Non-essential amino acids | Invitrogen | #11140-036 |
| OregonGreen ligand | Promega | #G2801 |
| PD0325901 | Axon Mechem | #1408 |
| Penicillin/streptomycin | Invitrogen | #15140-122 |

| Reagent/Resource | Reference or source | Identifier or catalogue number |
|---|---|---|
| Phenol red-free DMEM | Gibco | #31053028 |
| Poly-l-ornithine | Sigma-Aldrich | #P4957 |
| ProLong™ Gold Antifade Mountant | Invitrogen | #P36930 |
| Pulled glass capillary | Harvard Apparatus | #EC1 64-0766 |
| Puromycin dihydrochloride | Sigma-Aldrich | #P8833 |
| Q5 HotStart High Fidelity Polymerase | New England Biolabs | #M0493S |
| Sodium pyruvate | Invitrogen | #11360-039 |
| SYBR™ Green Nucleic Acid Gel Stain | Invitrogen | #S7563 |
| T4 DNA Ligase | New England Biolabs | #M0202S |
| Triton X-100 | Sigma-Aldrich | #X100 |
| Trypan blue | Gibco | #15250061 |
| Trypsin | Invitrogen | #15090-046 |
| Wheat Germ Agglutinin CF®488 A Conjugate (WGA488) | Biotium | #BT29022-1 |
| X-Gal | Promega | #V3941 |
| Zero Blunt TOPO PCR Cloning Kit | Invitrogen | #450245 |
| **Software** | | |
| R version 4.2.1 (tidyverse, rstatix) | (R Core Team, 2024) | |
| Bluues Version 2.0 | (Walsh et al, 2012) | |
| ColabFold | (Mirdita et al, 2022) | |
| ImageJ 1.53t (FIJI) | (Schindelin et al, 2012) | |
| **Other** | | |
| BD FACSAria II cell sorter | BD Biosciences | |
| BD LSRFortessa cell analyzer | BD Biosciences | |
| Celigo 4 Channel | Nexcelom Bioscience | |
| ECLIPSE Ti | Nikon | |
| Electro Square Porator ECM830 | BTX | |
| Opera Phenix™ High-Content Screening System | PerkinElmer | |
| SP8 Inverted confocal microscope | Leica | |
| TC20 automated cell counter | BioRad | |
| VT1000 S vibratome | Leica | |

## Construct design

We chose the following parts for the PUFFFIN construct design: the strong, ubiquitous CAG promoter (Niwa et al, 1991) should ensure high expression levels of the PUFFFIN construct and is followed by a Kozak consensus sequence (GCCGCCACC). The PUFFFIN fusion proteins start with a secretion signal peptide (s) from the human serum albumin (HSA) precursor (Dugaiczyk et al, 1982), optimised for mammalian expression by balanced GC

content and improved codon adaptation index (0.96 instead of 0.65 in original sequence), followed by a short two amino acid RG linker to maintain the sequence context of the HSA precursor, to prevent issues with cleavage of the signal peptide upon secretion. The sequence of +36GFP (Lawrence et al, 2007) is codon optimised for expression in mammalian cells (Bar-Shir et al, 2015), the N-terminal His-tag was removed from the original sequence. The 14 amino acid-long linker is glycine-serine rich for stability, while the small polar amino acid threonine makes it flexible (Chen et al, 2013). To increase the fluorescent signal of s36GFP, it was fused to an amplifier: either the ultra-bright green-fluorescent protein mNeonGreen (Shaner et al, 2013) or a HaloTag (Los et al, 2008). An internal ribosomal entry site (IRES) follows the s36GFP-amplifier coding sequence, to allow expression of a downstream puromycin N-acetyltransferase (PAC) gene, which confers resistance (PuroR) to the antibiotic puromycin dihydrochloride (de la Luna et al, 1988), enabling selection of clones with stable integration of the PUFFFIN constructs. The resistance gene can be replaced with a functional transgene of interest (GOI). A 2A self-cleaving peptide from porcine teschovirus-1 (p2A) (Tang et al, 2016) separates PAC and the subsequent fluorescent protein. The red-fluorescent mCherry (Shaner et al, 2004) was fused to three C-terminal SV40 nuclear localisation signals (3NLS) to drive nuclear localisation (Malaguti et al, 2013, 2022). Alternatively, cytoplasmic blue-fluorescent tagBFP (Subach et al, 2008) was used instead of mCherry-3NLS. A STOP cassette, consisting of a synthetic polyA (SPA) site (Levitt et al, 1989) and a C2MAZ terminator-binding sequence (Yonaha and Proudfoot, 2000) halts downstream transcription and translation. The whole expression construct is flanked by chicken β-globin insulators (cHS4) to prevent silencing at the random integration site (Chung et al, 1993). This was confirmed by culturing secretors to passage 15 without a decrease in s36GFP-mNG or mCherry-3NLS expression levels determined by flow cytometry. Ampicillin can be used for selective outgrowth of competent cells. We recommend linearisation with the PvuI restriction enzyme. These random integration plasmids containing s36GFP-mNeonGreen or s36GFP-HaloTag are 9913 bp and 10,159 bp in size, respectively.

For applications where a large plasmid size decreases delivery efficiency, we designed minimal PUFFFIN plasmids without insulators or mammalian selection gene, the smallest of which is 6840 bp in size (pyCAG-s36GFP-mNG-t2A-mCherry-3NLS-polyA). This minimal construct was adapted for the construction of a lentiviral vector by an in-house service following a standard protocol (McCloskey et al, 2014).

All plasmids created for this study will be deposited on Addgene and can be requested through the corresponding author.

## Protein folding and charge prediction

Protein folding prediction for +36GFP, s36GFP-mNG, and s36GFP-HT was done with ColabFold (Mirdita et al, 2022). Bluues Version 2.0 (Walsh et al, 2012) was used for prediction of theoretical net charge and electrostatic surface potential of the folded sequences and wild-type GFP (PDB ID: 4kw4).

## EMMA cloning: DNA domestication, assembly, and purification

In the original EMMA toolkit publication, the authors domesticate DNA parts by subcloning them into different entry vectors for each

position in the assembly (Martella et al, 2017). We modified this strategy in order to use the same entry vector for each position. PCR primers for DNA domestication were designed as follows: 5′-(N)$_6$-BsmBI binding site-N-fusion site-(N)$_{20}$-3′. As an example, the primers for position 1 were: Forward: 5′-(N)$_6$-CGTCTC-N-TAGG-(N)$_{20}$-3′, Reverse: 5′-(N)$_6$-CGTCTC-N-CCAT-(N)$_{20}$-3′. PCR amplification was performed with a Q5 HotStart High Fidelity Polymerase (NEB, #M0493S), and resulting blunt-ended PCR products were cloned into a pCR-Blunt II-TOPO vector (Invitrogen Zero Blunt TOPO PCR Cloning Kit; #450245) to generate domesticated DNA. This strategy allows for the creation of functional DNA sequences spanning multiple positions, without the need for assembly connectors. New DNA sequences, e.g. for the s36GFP-mNG fusion protein, were custom-synthesised by GeneArt or Integrated DNA Technologies.

TOPO Cloning reactions were transformed into NEB® 10-beta Competent E. coli (High Efficiency) (NEB, #C3019I) following the manufacturer's instructions. Bacteria were plated onto pre-warmed kanamycin resistance LB agar plates in the presence of X-gal (Promega, #V3941) for blue/white screening, and incubated overnight at 37 °C. White colonies were picked into 5 ml LB in the presence of kanamycin, and incubated at 37 °C prior to DNA purification.

EMMA assemblies of parts into the YCe3736_HC_Amp_ccdB_receiver vector (a kind gift from Prof Steven Pollard, also available on Addgene: #100637) were carried out using the NEBridge Golden Gate Assembly Kit (BsmBI-v2) (NEB, #E1602L), using 75 ng of each part and 2 μl of enzyme mix in a 20 μl reaction, performing 60 cycles of digestion (42 °C, 5 min) and ligation (16 °C, 5 min), followed by heat inactivation of enzymes (60 °C, 5 min). For assemblies involving domesticated parts harbouring extra BsmBI restriction sites, an extra ligation step was performed following heat inactivation: the reaction volume was increased to 30 μl, adding 400 U (1 μl) T4 DNA Ligase (NEB, #M0202S), and incubating the reactions at 16 °C for an additional 5–16 h.

About 10 μl of the assembly reaction was transformed into NEB® 5-alpha Competent E. coli (Subcloning Efficiency) (NEB, #C2988J) following the manufacturer's instructions. Bacteria were plated onto pre-warmed ampicillin resistance LB agar plates at two densities: 1/10 and 9/10 of the transformation reaction, and incubated overnight at 37 °C. Colonies were picked into 5 ml LB in the presence of ampicillin, and incubated at 37 °C prior to DNA purification. In our experience, colonies obtained with subcloning efficiency E. coli usually contain the correctly assembled plasmid. Should no colonies be obtained with subcloning efficiency E. coli, the remaining 10 μl of the assembly reaction can be transformed into high-efficiency E. coli. In our experience, in this instance, it is necessary to screen more bacterial colonies to identify correct assemblies.

Plasmid DNA was purified using the Monarch Plasmid DNA Miniprep Kit (NEB, #T1010), and subjected to diagnostic restriction digest and subsequent gel electrophoresis. Sequences of clones with correct banding patterns were verified by either Sanger sequencing of inserts or whole plasmid sequencing with Oxford Nanopore Technologies.

## Cell line generation and maintenance

HEK293 cells or E14Ju09 mouse embryonic stem cells (mESCs) were transfected with 3 μg of linearised plasmid using Lipofectamine 3000 Transfection Reagent (Invitrogen, #L3000001) in wells of a six-well plate. Selection with 2 μg/ml puromycin dihydrochloride (Sigma-Aldrich; #P8833) was started 48 h after transfection for transfected cells and a mock transfection control. Cells were replated from the six-well plate at 1/10 and 9/10 densities onto 10 cm cell culture dishes and resulting colonies picked and expanded as monoclonal lines. Polyclonal lines were considered to only contain stable integrants following the death of all cells in the mock transfection control dishes.

All HEK293-derived cell lines were maintained in HEK293 culture medium—Dulbecco's Modified Eagle Medium (DMEM, high glucose, pyruvate; Gibco, #41966029) with 10% foetal calf serum (FCS; Biosera, #FB-1280/500)—under standard culture conditions (37 °C and 5% $CO_2$) and passaged after washing with Dulbecco's phosphate-buffered saline (PBS; Sigma-Aldrich, #D8537) using 0.025% trypsin (Invitrogen, #15090-046). To enrich the stable polyclonal s36GFP-HT secretor line for high transgene expressing cells, cells were prepared for flow cytometry as described below, and sorted based on fluorescent marker expression on a BD FACSAria II. After sorting, cells were cultured for 2 weeks in a culture medium supplemented with penicillin/streptomycin (100 U/ml; Invitrogen, #15140-122).

The mESC cell lines were routinely cultured in LIF/Serum + PD03 medium on gelatine coating (0.1%; Sigma-Aldrich, #G1890): Glasgow Minimum Essential Medium (GMEM; Gibco, #1171035) supplemented with 10% foetal calf serum (FCS; Biosera, #FB-1280/500), 100 U/ml LIF (produced in-house), 2-mercaptoethanol (100 nM; Life Tech, #31350010), 1X non-essential amino acids (Invitrogen, #11140-036), sodium pyruvate (1 mM; Invitrogen, #11360-039), and PD0325901 (0.5 μM; Axon Medchem, #1408). Stable lines were cultured for a minimum of 14 days in 2iLIF on fibronectin-coating (Sigma-Aldrich, #F1141) at the start of experiments. 2iLIF was made with 50% DMEM/F-12 (Gibco, #21331020) and 50% Neurobasal Medium (Gibco, #21103049), supplemented with 2-mercaptoethanol (100 nM, Gibco, #31350010), 1X non-essential amino acids (Invitrogen, #11140-036), L-glutamine (2 mM; Invitrogen, #25030-024), 1X N2 (Gibco, #17502048), 1X B27 (Gibco, #17504044), LIF (100 U/ml; produced in-house), CHIR99021 (3 μM; Axon Medchem, #1386), and PD0325901 (1 μM; Axon Medchem, #1408). 2i (2iLIF without LIF) and N2B27 (2iLIF without LIF, CHIR, and PD03) medium were used in addition to 2iLIF as indicated for experiments. Cells were cultured under standard culture conditions and passaged after washing with Dulbecco's phosphate-buffered saline (PBS; Sigma-Aldrich, #D8537) using Accutase (Invitrogen, #00-4555-56).

All routinely used cell lines tested negative for mycoplasma contamination.

## Co-culture experiments

Secretors and unmodified cells were detached from the cell culture vessel using trypsin or Accutase, resuspended in culture medium, pelleted by centrifuging at $300 \times g$ for 3 min, resuspended in an appropriate amount of medium and counted to plate cells at unmodified cells-to-secretor ratios indicated for each experiment. Co-culture cell suspensions were thoroughly mixed before plating in the respective vessel. Single cultures for secretors and unmodified cells were included in all experiments.

## Flow cytometry

For flow cytometry experiments, for 2–48 h time points cells were cultured under standard conditions in 12-well plates, detached using trypsin and quenched in HEK293 culture medium; for 15–60 min time points cells were mixed in suspension and pelleted at $300 \times g$ for 1 min prior to incubation at 37 °C and 5% $CO_2$; or for 0 h time points mixed on ice immediately prior to analysis.

For pelleted suspension cultures with different cell lines as unmodified cells, mShef7 human pluripotent cells, E14tg2a mouse pluripotent cells and UCH1N human chordoma cells were received as fresh single cell suspensions and immediately counted for 1:1 co-culture ratio. For co-cultures unmodified cells were mixed with HEK293 secretors in HEK293 culture medium (500 μL total volume), for single cultures cells remained in their respective medium. All cell suspensions were pelleted at $300 \times g$ for 1 min prior to incubation for 90 min at 37 °C and 5% $CO_2$.

Cells were pelleted by centrifuging at $300 \times g$ for 3 min and resuspended in 10% FCS in PBS with either DAPI (1 μM; Biotium, #40043) or DRAQ7 (300 nM; Abcam, #ab109202) live/dead staining. No DAPI/No DRAQ7 controls were included in all experiments. Samples were analysed on a BD LSRFortessa Cell Analyzer using V 450/50-A (blue fluorescence), B 530/30-A (green fluorescence), Y/G 610/20-A (red fluorescence) and R 670/14-A (far-red fluorescence) laser/filters combinations.

## Live imaging

Cells were plated on 7.5 μg/ml fibronectin-coated (Sigma-Aldrich, #F1141) 4-well μ-slides (Ibidi, #80426) with or without removable culture-inserts (Ibidi, two-well #80209 or three-well #80369). Culture inserts were removed 8 h prior to imaging and HEK293 culture medium was changed to imaging medium—phenol red-free DMEM (Gibco, #31053028) with 10% FCS (Biosera, #FB-1280/500) and 2% glutamate/pyruvate (100 mM sodium pyruvate, Invitrogen #11360-039; 200 mM L-glutamine, Invitrogen, #25030-024). Slides were imaged for brightfield, red and green fluorescence at 15 min intervals over 19 h on an Opera Phenix™ High-Content Screening System (PerkinElmer) under standard culture conditions.

## Quantification of labelling distance from live imaging data

For the final time point of the live imaging data (19 h), an 1890-μm-long border between secretors and unmodified cells was analysed. In FIJI (Schindelin et al, 2012), a 10 μm × 300 μm rectangle was placed orthogonally to the border starting with the last secretor and its intensity profile for green and red fluorescence plotted (Analyze - Plot profile). The rectangle was then moved 10 μm along the border and adjusted to the last secretor. This was repeated for the whole length of the border resulting in 189 profiles for GFP and mCherry. Fluorescence intensity values were normalised to 1, the mean of all profiles and the 95% confidence interval of the mean of GFP and mCherry, and the half-maximum of GFP were calculated and visualised. In the same image, the diameter for 200 representative cells with clear outlines in brightfield was measured, with the mean and its 95% confidence interval calculated.

## HaloTag staining and analysis

Cells were plated for co-culture experiments as described above, either in 12-well plates for flow cytometry or in fibronectin-coated eight-well μ-slides for live imaging. On the day of analysis, the HEK293 culture medium was replaced by imaging medium containing either Janelia Fluor 646 (JF646) ligand (10 nM; Promega, #GA1120) or OregonGreen ligand (50 nM; Promega, #G2801) or no HaloTag dye. Cells were incubated for 2 h at 37 °C and prepared for and processed by flow cytometry as described above. Hoechst 33342 (100 ng/ml; Apexbio, #A3472-APE) nuclear staining was added to the medium of the eight-well slide 1 h prior to imaging, and WGA488 (1 μg/ml; Wheat Germ Agglutinin CF®488 A Conjugate; Biotium, #BT29022-1) membrane staining was added to single wells immediately before imaging on an Opera Phenix™ High-Content Screening System (PerkinElmer).

## Colony-forming efficiency in clonal commitment assay

### Culture and fluorescence-activated cell sorting

mESC cell lines were generally cultured in 2iLIF. Adapting an existing protocol for testing the colony-forming efficiency during exit from naïve pluripotency (Kalkan et al, 2017), four conditions were tested: (1) Unmodified cells and secretors were cultured in 2iLIF prior to co-culture in 2iLIF. (2) Unmodified cells and secretors were cultured in 2i for 24 h prior to co-culture in N2B27. (3) Unmodified cells were cultured in 2i for 24 h and secretors in 2i for 24 h followed by N2B27 for 24 h prior to co-culture in N2B27. (4) Unmodified cells were cultured in 2i for 24 h and secretors in 2i for 24 h followed by N2B27 for 48 h prior to co-culture in N2B27. 2iLIF single cultures for unmodified cells and secretors were included as controls.

All co-cultures were seeded with a 9:1 unmodified cell : secretor ratio at a density of $5 \times 10^4$ cells per cm$^{-2}$ for a 24 h co-culture. Culture medium was replaced with medium (2iLIF or N2B27, respectively) containing JF646 ligand (10 nM; Promega, #GA1120) 1.5 h prior to sample processing. Co-cultures were washed with PBS, detached with accutase and quenched in respective medium, and pelleted by centrifuging at 300 g for 3 min. Pellets were resuspended in either 2iLIF + DAPI or N2B27 + DAPI (1 μM DAPI; Biotium, #40043). All samples were kept on ice from this point onwards.

Cells were sorted using a BD FACSAria II Cell Sorter. Gating for secretors, labelled unmodified cells ('neighbours'), and unlabelled unmodified cells ('non-neighbours') was determined using single culture controls. For the clonal commitment assay, 20,000 cells were sorted into 200 μL 2iLIF + Pen/Strep (Invitrogen, #15140-122).

For the clonal commitment assay, sorted cell samples were counted using the BioRad TC20 Automated Cell Counter and trypan blue live/dead staining. For each sample (secretors, neighbours, and non-neighbours for four different conditions with three technical replicates per sample) 250 live cells were plated in 2iLIF + Pen/Strep into one well of a six-well plate with poly-l-ornithine (Sigma-Aldrich, #P4957) and laminin (Sigma-Aldrich, #L2020) coating. Plates were cultured for 6 days under standard conditions and half media changes were done every other day. On the day of analysis (1 μg/ml; Wheat Germ Agglutinin CF®488 A

Conjugate; Biotium, #BT29022-1) membrane staining was added to the medium and plates imaged on the Celigo plate imager. Colonies were counted in FIJI (Schindelin et al, 2012).

### Statistical analyses for colony-forming efficiency

Data analysis was performed in R version 4.2.1 (R Core Team, 2024) using the packages tidyverse and rstatix. For the colony-forming efficiency (CFE), the mean colony number and the standard error of the mean (SEM) were calculated for neighbours and non-neighbours in all culture conditions. Percentages relative to populations in 2iLIF were calculated using the mean colony numbers. *P* values for neighbour-to-non-neighbour ratios represent Bonferroni-adjusted, two-sided Fisher's exact tests relative to the 2iLIF medium, using the sum of the colony counts averaged over five independent biological replicates.

## Antibody staining and immunofluorescence microscopy of mESCs

Co-cultures and single cultures of mESC secretors and unmodified cells were cultured on 7.5 µg/ml fibronectin-coated (Sigma-Aldrich, #F1141) eight-well µ-slides (Ibidi, #80806-90) for 24 h under standard conditions. Cells were washed with PBS and fixed for 20 min at RT with 4% formaldehyde in PBS. After three washes with PBS, slides were incubated overnight at 4 °C with blocking solution: PBS with 5% donkey serum and 0.1% Triton X-100). Primary antibodies diluted in blocking solution were added and the slides were incubated for 3 h at RT, followed by washing three times with PBS. The slides were incubated with secondary antibodies and DAPI diluted in a blocking solution for 1 h at RT. Slides were washed three times with PBS before adding 90% glycerol in PBS to all wells. Slides were kept at 4 °C until imaged on a Nikon ECLIPSE Ti or Leica SP8 inverted confocal microscope.

## Brain slice experiments

### Animals

All animal experiments were carried out in accordance with the guidelines established by the Animal Care (Scientific Procedures) Act 1986 and under the authority of Home Office project license PP5764018 (L.Z.).

### Mouse organotypic brain slice cultures

This brain slice culture protocol was previously published (Eigel et al, 2019). Briefly, C57BL/6J mouse pups (postnatal days 2–4) were decapitated and their brains were dissected in cold Hibernate™-A Medium (Gibco, #A1247501) on ice. Dissected brains were then mounted on a vibratome (Leica VT1000 S) and 250-300 µm coronal cortical slices were collected in ice-cold Hibernate-A medium. The slices were immediately transferred onto Millicell cell culture inserts (30 mm, hydrophilic PTFE, 0.4 µm, Merck-Millipore, #PICM0RG50) with pre-warmed slice medium containing 50% MEM (Gibco, #12360038) with 25% Earle's Balanced Salt Solution (Gibco, #24010043), 25% heat-inactivated horse serum, New Zealand origin (Gibco, #26050088), 1% Glutamax™ Supplement (Thermo Scientific, #35050-038), 1% penicillin–streptomycin (Invitrogen, #15140-122), 0.5% Amphotericin B (Gibco, #15290026) and 6.5 mg/ml glucose (Sigma-Aldrich, #G8769). Each membrane insert contained three cortical slices

(1 slice/mouse). The culture medium was changed every other day. Slices were kept in culture for 7 days at 37 °C and 5% $CO_2$ prior to lentiviral transduction. $8.9 \times 10^6$ TU/ml of PUFFFIN lentivirus was added to the slices which were cultured for a further 10 days after viral transduction.

### Slice culture immunofluorescence

Cortical slices were washed once with PBS before they were fixed with 4% paraformaldehyde (PFA) in PBS for 1 h at room temperature (RT). The slices were subsequently rinsed in PBS and blocked in 3% heat-inactivated horse serum heat-inactivated horse serum, New Zealand origin (Gibco, #26050088), 2% Bovine Serum Albumin (Sigma-Aldrich, #A7906) and 0.5% Triton X-100 (Sigma-Aldrich, #X100) in PBS for 2 h at RT. Following blocking, the slices were incubated for 48 h with primary antibodies at 4 °C on a shaker. The slices were then washed three times with blocking solution and incubated with the appropriate secondary antibodies overnight at 4 °C on a shaker. Finally, the slices were washed in PBS, counterstained with Hoechst 33342 solution (Thermo Scientific, #62249) and mounted on a glass microscopic slide using Fluoromount-G® mounting medium (SouthernBiotech, #0100-01).

Primary antibodies used: Rabbit- anti-RFP, pre-adsorbed (RRID:AB_2209751, Rockland, #600-401-379, 1:1000), and mouse anti-mNeonGreen (32F6, RRID: AB_2827566, Proteintech, #17882377). Secondary antibodies were all purchased from Thermo Fisher Scientific (Alexa, 1:1000). Images were obtained using confocal laser scanning microscopy (Leica microsystems, TCS SP8) and processed using Fiji image processing package (Schindelin et al, 2012).

## Chick experiments

### Chicken embryos and electroporation

Fertilised chicken embryos (Gallus gallus) were obtained from Henry Stewart & Co. Ltd and kept in humidified incubators at 38 °C until the desired stage (Hamburger and Hamilton, 1951). Eggs were incubated for 35 h to reach HH10 stages and then injected in the caudal lateral epiblast via a pulled glass capillary (Harvard Apparatus, #EC1 64-0766) with plasmid solution mixed with SYBR green (Invitrogen, #S7563). PUFFHalo minimal plasmid was injected at a final concentration of 250 ng/µl to achieve mosaic labelling.

Ex ovo electroporations were performed similarly to those previously described (Williams and Sauka-Spengler, 2021). In short, embryos were isolated via hole-punched Whatman filter paper and placed ventral side up in an electroporation chamber with a negative electrode at the bottom filled with Tyrodes saline. The plasmid mixture was injected in the space between the vitelline membrane and the CLE of the embryo. The positive electrode was placed over the CLE to achieve targeted plasmid electroporation (pulse regime—7 V, 50 ms. length, 3 pulses, 500 ms. interval). Embryos were then cultured for 24 h using the EC culture method (Chapman et al, 2001) in 35 mm dishes and screened for transfection efficiency (i.e. presence of mCherry fluorescence within the trunk). On the day of analysis, embryos were incubated in an L15 medium containing Janelia Fluor 646 (JF646) ligand (500 nM; Promega, #GA1120) for 1 h at 37 °C.

Healthy and transfected embryos were dissected and processed for immunohistochemistry or flow cytometry.

*Immunohistochemistry of chick embryos*

Chicken embryos were fixed in 4% paraformaldehyde for 1 h at RT. Embryos were washed twice in PBS, incubated with 10 µg/ml DAPI and 1:500 RFP-Booster Alexa Fluor® 568 (Chromotek, #rb2AF568) for 2 h at RT and washed extensively in PBS. Embryos were mounted between coverslips with ProLong™ Gold Antifade Mountant (Invitrogen, #P36930) and imaged with a Leica SP8 confocal microscope. Images were processed with Fiji (Schindelin et al, 2012).

*Flow cytometry of chick cells*

For flow cytometry experiments, embryo trunks were dissected, and 3–5 trunks were pooled per sample. Samples were dissociated in FACSmax Cell Dissociation Solution (Amsbio, #AMS.T200100) for 20 min at 37 °C with shaking. Cells were pelleted by centrifuging at $600 \times g$ for 4 min and resuspended in 1% BSA in PBS with DAPI (1 µg/ml). Samples were analysed on a BD LSRFortessa Cell Analyzer using V 450/50-A (blue fluorescence), Y/G 610/30-A (red fluorescence), and R 670/30-A (far-red fluorescence) laser/filters combinations.

## Data availability

The datasets produced in this study (imaging files, flow cytometry files and numerical data) are available alongside a README file at the Figshare data repository: https://doi.org/10.6084/m9.figshare.25745106.

The source data of this paper are collected in the following database record: biostudies:S-SCDT-10_1038-S44318-024-00154-w.

## Peer review information

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

## Acknowledgements

We would like to thank Mihaly Badonyi for support with construct design of the s36GFP label and the s36GFP-mNG and PUFFHalo fusion proteins, as well as advice on statistical analyses. We thank Charles AC Williams for help with HaloTag staining and imaging support. We are thankful to Rachel White and Steven M Pollard for providing the YCe3736_HC_Amp_ccdB receiver vector and assistance with EMMA cloning. We thank the labs of Val Wilson and Guillaume Blin for the valuable feedback and reagents shared, especially Heather MacPherson and Raffee Wright II, for providing MShef7 and UCH1N cell suspensions. We are grateful to Pamela Brown for the construction and production of the PUFFFIN lentivirus. We appreciate the help of Linh Huynh and Keisuke Kaji with high-throughput imaging of the colony formation assay. We thank Miguel Robles Garcia for facilitating micropatterning of PUFFFIN cells to produce our cover image. Thank you to the group members of the Lowell Lab, especially but without any particular order, to Jen Annoh, Sophie Brumm, Eleanor Earp, Aisling Fairweather, Matthew French, Alicia Louis Perez Lezcano, and Maria Rosa Portero Migueles. We are grateful to the facilities at the Institute for Regeneration and Repair of the University of Edinburgh, especially Fiona Rossi at the Flow Cytometry Core Facility, Justyna Cholewa-Waclaw at the High-Content Screening Facility, Matthieu Vermeren at the Imaging Facility, and Theresa O'Connor at the Tissue Culture Facility; and Iain Porter and Adrian Garcia Burgos from the IMPACT Facility at the Centre for Discovery Brain Sciences. TL was supported by the Integrative Cell Mechanisms PhD Programme, with funding from Wellcome [218470], and the University of Edinburgh School of Biological Sciences and a core grant to the Wellcome Centre for Cell Biology [203149]. MM was supported by a BBSRC Engineering Biology Transition Award (BB/W014610/1), and by a University of Edinburgh School of Biological Sciences internal Seeding Fund award. SL is supported by a Wellcome Trust Senior Fellowship [220298]. LZ is supported by a University of Edinburgh Chancellor's Fellowship and by the Simons Initiative for the Developing Brain. GLMB was supported by EMBO ALTF (792-2021) and UKRI (EP/X031225/1).

## Author contributions

**Tamina Lebek**: Conceptualisation; Resources; Data curation; Formal analysis; Funding acquisition; Validation; Investigation; Visualisation; Methodology; Writing—original draft; Writing—review and editing. **Mattias Malaguti**: Supervision; Investigation; Visualisation; Methodology; Resources; Funding acquisition. Writing—review and editing. **Giulia LM Boezio**: Data curation; Validation; Investigation; Visualisation; Methodology; Writing—review and editing. **Lida Zoupi**: Data curation; Validation; Investigation; Visualisation; Methodology; Writing—review and editing. **James Briscoe**: Supervision; Funding acquisition; Project administration; Writing—review and editing. **Alistair Elfick**: Resources; Supervision; Writing—review and editing. **Sally Lowell**: Supervision; Funding acquisition; Methodology; Writing—original draft; Project administration; Writing—review and editing.

Source data underlying figure panels in this paper may have individual authorship assigned. Where available, figure panel/source data authorship is listed in the following database record: biostudies:S-SCDT-10_1038-S44318-024-00154-w.

## Disclosure and competing interests statement

The authors declare no competing interests.

# Expanded View Figures

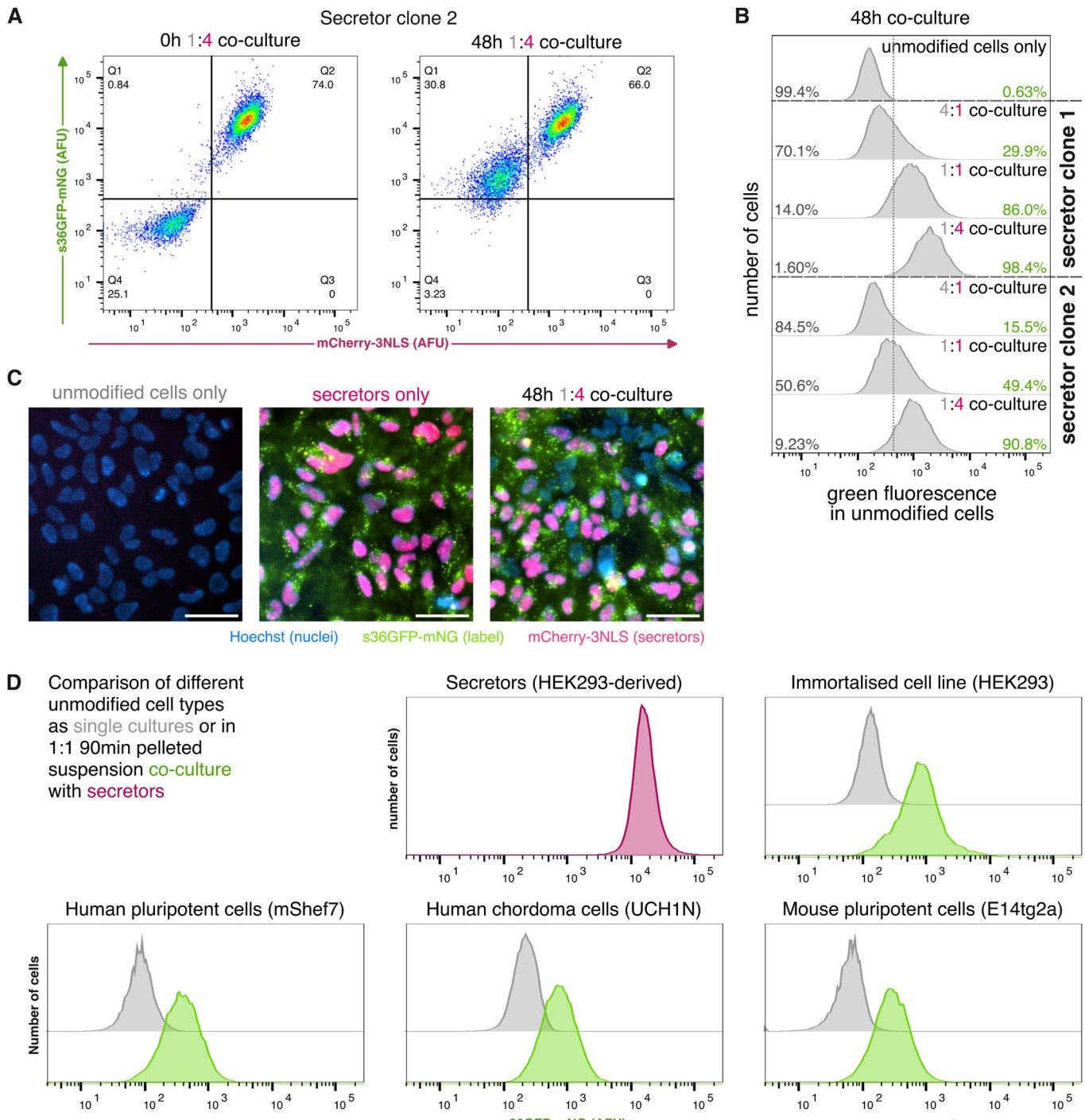

**Figure EV1.  Secretors can effectively label unmodified cells by transferring s36GFP-mNG.**

(A) Flow cytometry of a 1:4 co-culture of unmodified cells and a second independent monoclonal secretor line (secretor clone 2) at 0 and 48 h time points. Three independent experiments were performed, and a representative set is shown. (B) Flow cytometry of a 48 h co-culture with unmodified cells and either secretor clone 1 or secretor clone 2 seeded at different co-culture ratios. Grey numbers (bottom left of each plot) are the percentage of unmodified cells in Q4, and green numbers (bottom right of each plot) are the percentage of unmodified cells in Q1. Three independent experiments were performed, and a representative set is shown. (C) Live imaging of single cultures of unmodified cells or secretors and a 48 h 1:4 co-culture of unmodified cells and secretors with Hoechst nuclear staining. The scale bar is 50 μm. (D) Flow cytometry of 1:1 90 min pelleted suspension co-cultures of different unmodified cell types and HEK293-derived secretors.

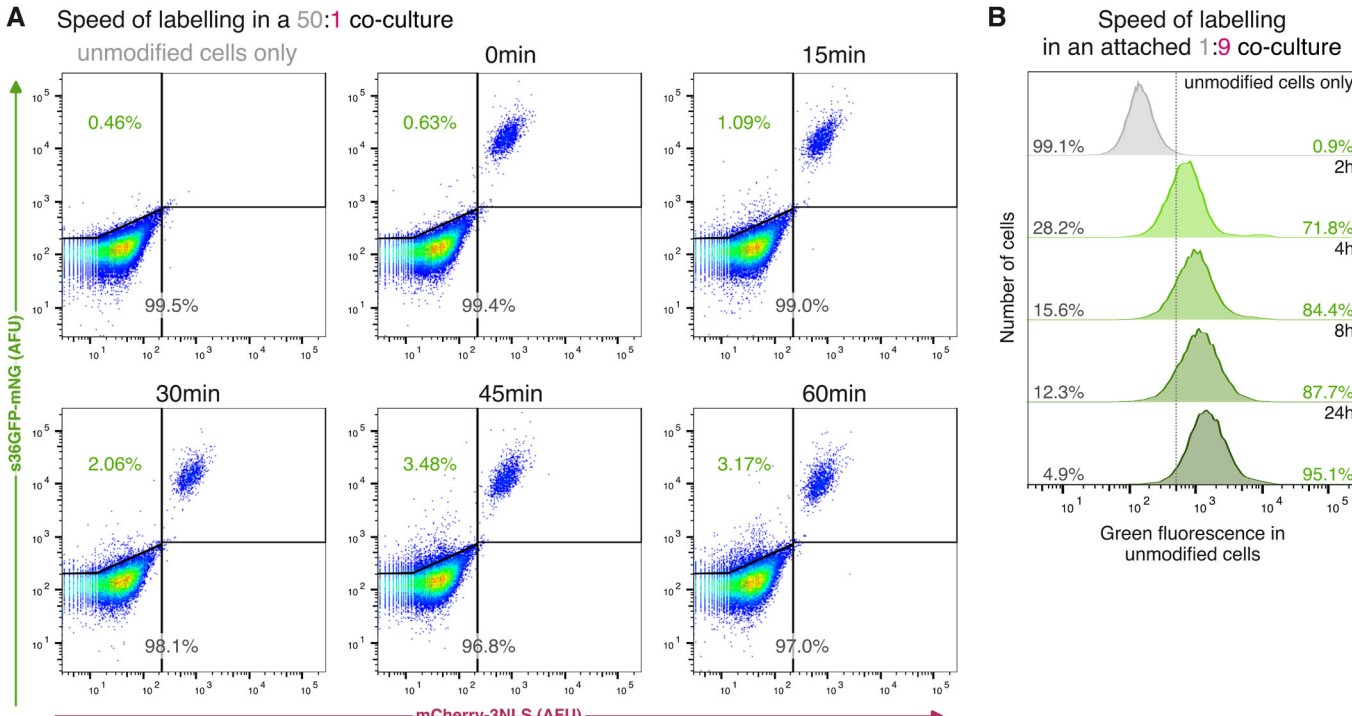

**Figure EV2.  PUFFFIN labelling is time-dependent and fast.**

(A) Speed of labelling with few secretors by flow cytometry analysis of different 50:1 unmodified cell:secretor ratios in a pelleted suspension co-culture as a time course experiment. Grey numbers are percentage of unmodified cells in top polygon, green numbers are percentage of unmodified cells in bottom polygon. 50,000 cells were analysed for each sample. Three independent experiments were performed, and a representative set is shown. (B) Speed of labelling in an attached monolayer co-culture shown by flow cytometry analysis of a 1:9 unmodified cell:secretor co-culture time course experiment. 10,000 unmodified cells were analysed for each sample. Grey numbers (bottom left of each plot) are the percentage of unmodified cells in Q4, and green numbers (bottom right of each plot) are the percentage of unmodified cells in Q1. Three independent experiments were performed, and a representative set is shown.

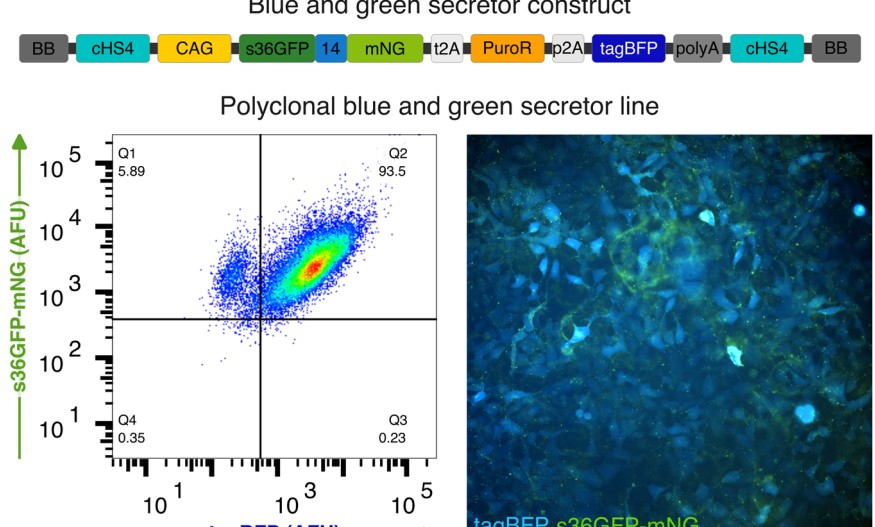

**Figure EV3. Modularity of the PUFFFIN expression construct allows change of colour for the secretor-cell-marker.**

Polyclonal secretors expressing s36GFP-mNG and cytoplasmic tagBFP made by random integration of the blue-green secretor construct (top) are shown by flow cytometry (left) and imaging live cells for green and blue fluorescence (right). The scale bar is 50 μm.

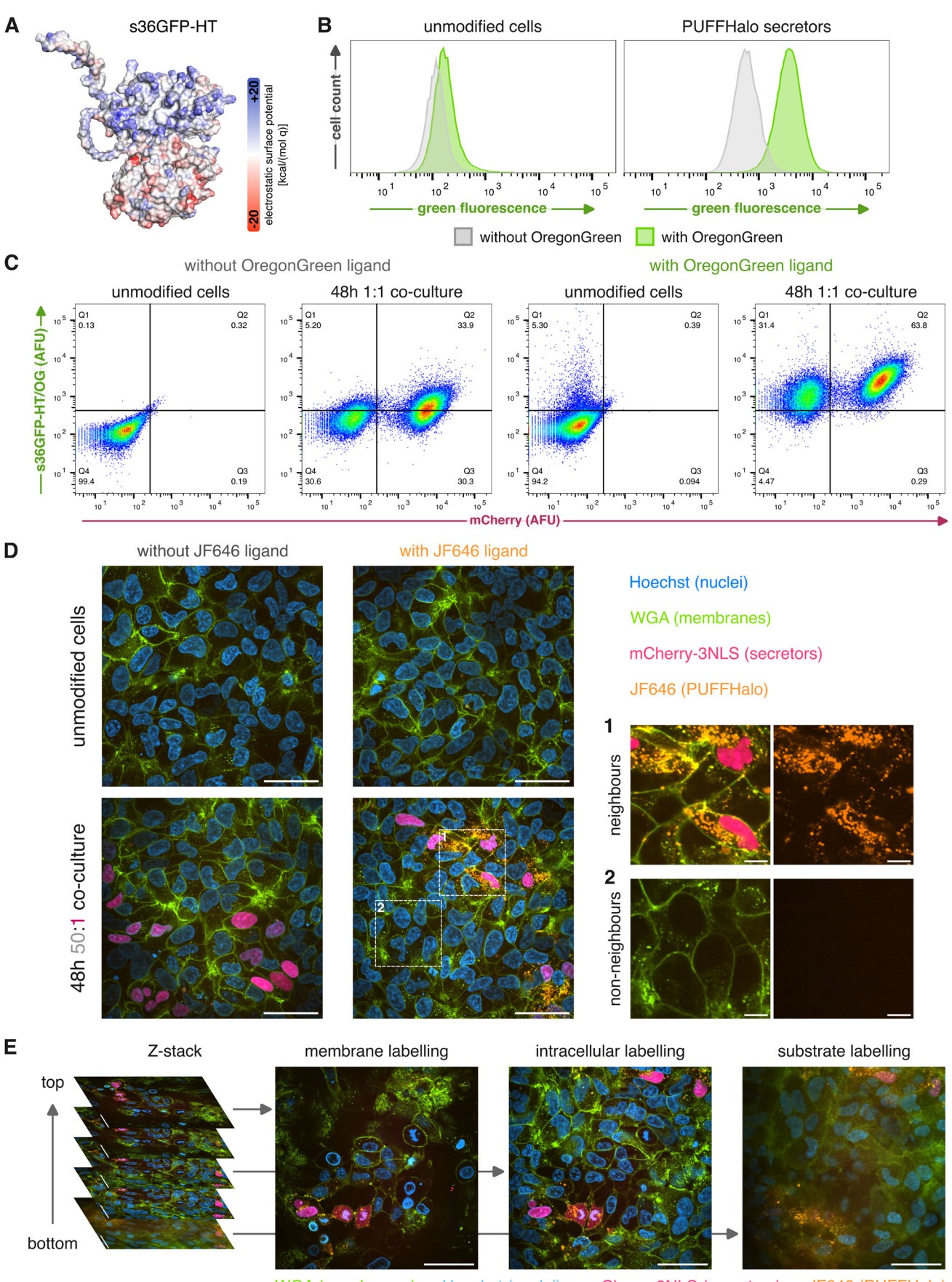

◀ **Figure EV4. PUFFHalo integrates HaloTag technology with PUFFFIN for choice-of-colour labelling.**

(**A**) Electrostatic surface potential is shown for s36GFP-HaloTag (AlphaFold predicted structure). (**B**) Unmodified cells and PUFFHalo secretors were incubated for 2 h with or without the HaloTag ligand OregonGreen. Three independent experiments were performed, and a representative set is shown. (**C**) Flow cytometry of 48 h 1:1 co-cultures of unmodified cells and PUFFHalo secretors were incubated for 2 h with or without the HaloTag ligand OregonGreen. Three independent experiments were performed, and a representative set is shown. (**D**) Live imaging of single cultures of unmodified cells and 48 h 50:1 co-cultures of unmodified cells and PUFFHalo secretors with or without the HaloTag ligand JF646, all with Hoechst nuclear staining and WGA488 membrane staining. Scale bars are 50 μm. Two regions of the 48 h 1:1 co-culture with JF646 are magnified to show neighbours and non-neighbours. Scale bars of (1) and (2) are 10 μm. (**E**) Three single planes of a Z-stack for a 48 h 1:1 co-culture with JF646, WGA488, and Hoechst staining were selected to highlight different localisations of the s36GFP-HT label. Scale bars are 50 μm.

**A**  Lentiviral minimal PUFFFIN expression construct

BB | CMV | s36GFP | 14 | mNG | t2A | mCherry-3NLS | oPRE | BB

**B**  Experimental design: Mouse cortical brain slices injected with PUFFFIN lentivirus

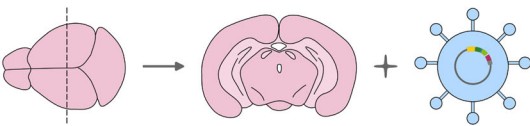

**C**  Mouse cortical brain slices 10 days after injection ± PUFFFIN lentivirus

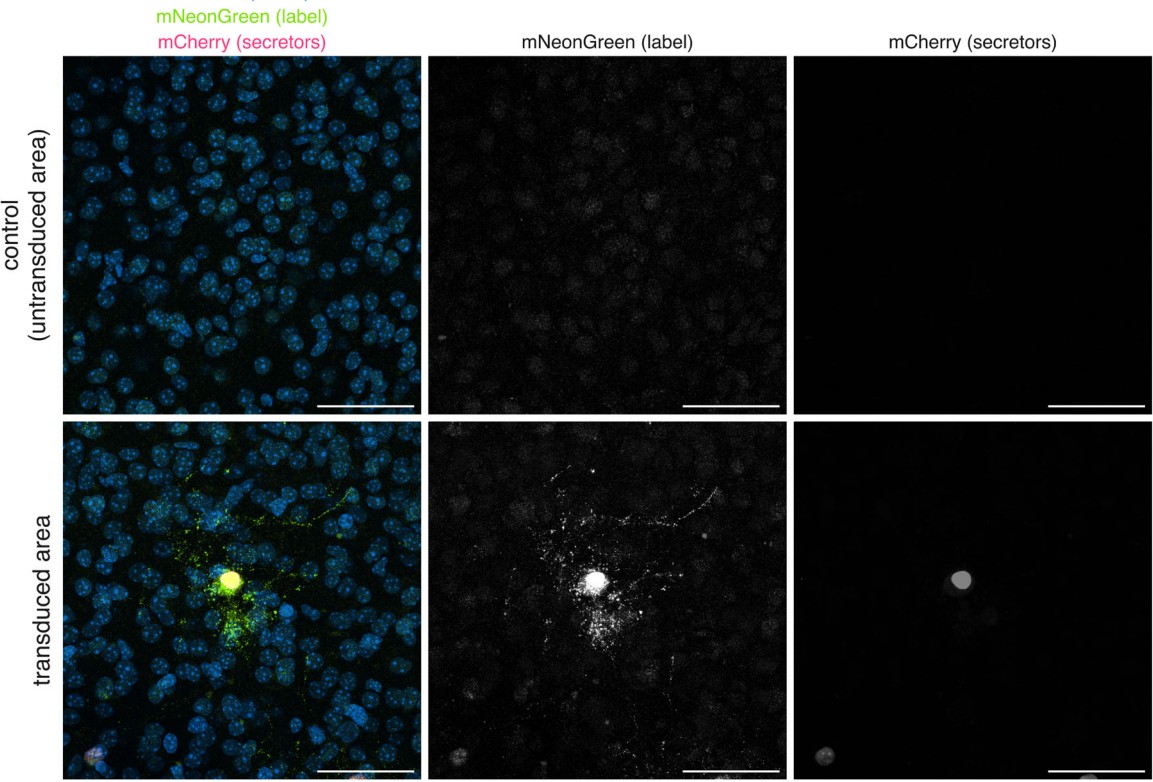

**Figure EV5.  PUFFFIN expression in mouse organotypic brain slices using a lentiviral vector.**

(**A**) Minimal PUFFFIN expression construct as used in lentivirus vector. (**B**) Illustration of the experimental design for transducing cultured mouse cortical slices with the PUFFFIN lentivirus. (**C**) Immunofluorescence of mouse cortical organotypic slices untransduced or transduced with 8.9 × 10⁶ TU/ml of PUFFFIN lentivirus, cultured for 10 days after injection, and stained for mNeonGreen and mCherry, and counterstained with Hoechst. Scale bars are 50 μm.

**A**  PUFFHalo labelling in 24h
9:1 co-cultures with naïve or
pre-differentiated secretors

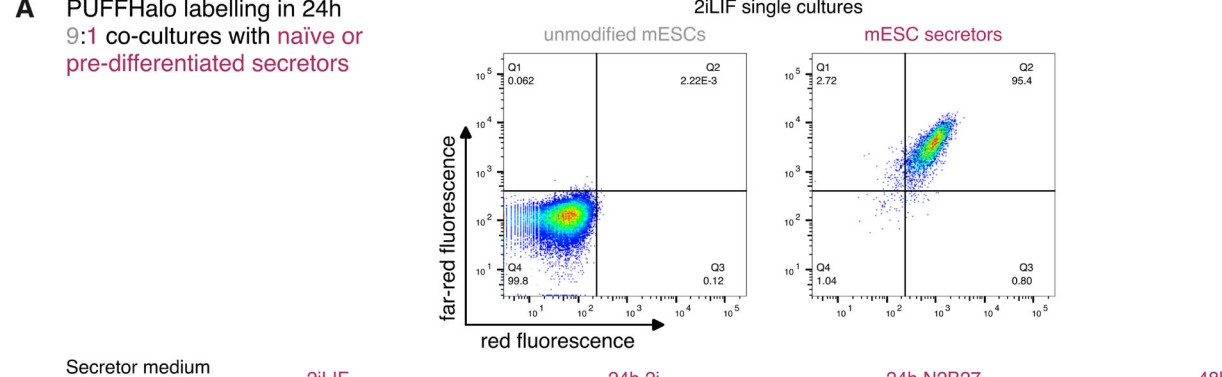

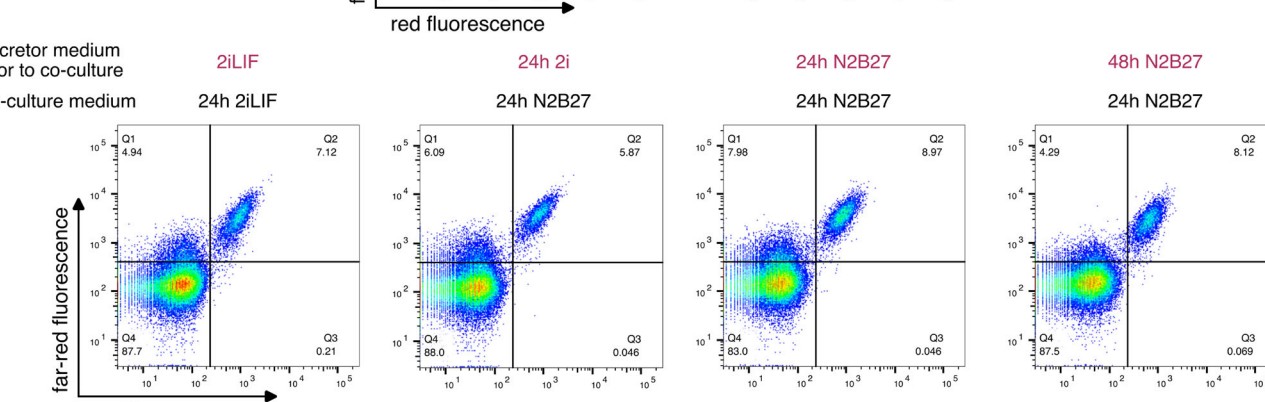

**B**  Clonal commitment assay: colonies counted after 6d in 2iLIF

Number of colonies = number of uncommitted naïve cells

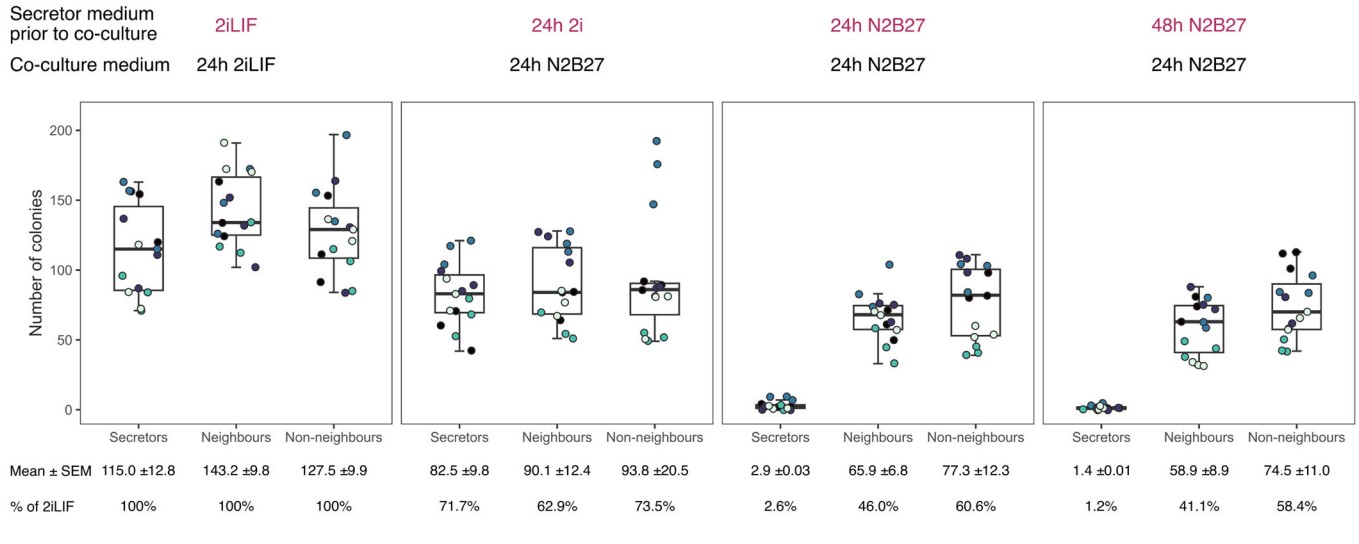

| | | 2iLIF | | | 24h 2i | | | 24h N2B27 | | | 48h N2B27 | |
|---|---|---|---|---|---|---|---|---|---|---|---|---|---|
| | | Secretors | Neighbours | Non-neighbours | Secretors | Neighbours | Non-neighbours | Secretors | Neighbours | Non-neighbours | Secretors | Neighbours | Non-neighbours |
| Mean ± SEM | | 115.0 ±12.8 | 143.2 ±9.8 | 127.5 ±9.9 | 82.5 ±9.8 | 90.1 ±12.4 | 93.8 ±20.5 | 2.9 ±0.03 | 65.9 ±6.8 | 77.3 ±12.3 | 1.4 ±0.01 | 58.9 ±8.9 | 74.5 ±11.0 |
| % of 2iLIF | | 100% | 100% | 100% | 71.7% | 62.9% | 73.5% | 2.6% | 46.0% | 60.6% | 1.2% | 41.1% | 58.4% |

Biol. replicate  ● 1  ● 2  ● 3  ● 4  ○ 5

**Figure EV6. PUFFHalo labelling in mESCs can be utilised for investigating exit from pluripotency.**

(**A**) PUFFHalo labelling shown for 2iLIF single cultures and 24 h 9:1 co-cultures in different media conditions shown by flow cytometry. Three independent experiments were performed, and a representative set is shown. (**B**) Colony formation efficiency (CFE) is shown as the number of colonies for the clonal commitment assay counted after 6 days in 2iLIF, coloured by biological replicate. Boxes denote data within the 25th and 75th percentiles, the middle line represents the median, and the whiskers extend from the upper and lower quartiles to a distance of 1.5 times from the median. Below the plot, mean colony numbers ± standard error of the mean (SEM) and percentages relative to 2iLIF are reported for secretors, neighbours and non-neighbours. Five independent experiments with three non-independent replicates were performed.

