## [Peer Review File · The EMBO Journal]

PUFFFIN: An ultra-bright, customizable, single-plasmid system for labelling cell neighborhoods

Tamina Lebek, Mattias Malaguti, Giulia Boezio, Lida Zoupi, James Briscoe, Alistair Elfick, and Sally Lowell

Corresponding author: Sally Lowell (sally.lowell@ed.ac.uk)

Review Timeline:

Transferred from Review Commons:	28th Oct 23
Editorial Decision:	6th Nov 23
Revision Received:	2nd May 24
Editorial Decision:	25th May 24
Revision Received:	5th Jun 24
Accepted:	10th Jun 24

Editor: Kelly Anderson

Transaction Report:

This manuscript was transferred to The EMBO Journal following peer review at Review Commons.

Review #1

1. Evidence, reproducibility and clarity:

Evidence, reproducibility and clarity (Required)

This manuscript reports the development of a new bright fluorescent reporter that allows to label neighbouring cells. The system is based on upon secretion and uptake of s36GFP, a positively supercharged fluorescent protein. The authors also develop a Halo tag that will allow for straight forward colour exchange, as well as a customisable single plasmid construct with modular components.

There are some minor suggestions that the authors may want to consider:

1. The authors conclude that "PUFFIN labelling is transferred rapidly between cells within minutes". They report in their time lapse experiments (Figure 2A,C) that sGFP can be detected within neighbours of secretors after 30 minutes when the cells are plated in a 50:1 non-labelled/secretor cell ratio, whereas it can be detected after 15 minutes when the cells are plated in a 1:9 ratio. Is there any synergistic effect on the signal when the proportion of secretors is increased or is this difference just because there is more signal, making it easier to visualise.
2. Is there any reason why the main Figure legends lack a title, but the supplementary figures have one?
3. In Figure 3, it may be helpful to label each option as A, B, C..
4. In Figure 4E, the legends + JF646 and -JF646 may be switched around. Shouldn't the orange box should be (+) and the grey box should be (-)?

2. Significance:

Significance (Required)

This is a very valuable tool to address how cells change the behaviour of those in their environment. It will be very valuable for those interested in cell non-autonomous effects within a cell population or tissue. It will be especially valuable for live cell imaging; pulse chase experiments as well as omics approaches to understand cell behaviour in niches.

3. How much time do you estimate the authors will need to complete the suggested revisions:

Estimated time to Complete Revisions (Required)

(Decision Recommendation)

Less than 1 month

Yes

Review #2

1. Evidence, reproducibility and clarity:

Evidence, reproducibility and clarity (Required)

The authors describe a new method, Positive Ultra-bright Fluorescent Fusion For Identifying Neighbours (PUFFFIN), to label with Fluorescent Proteins, neighboring cells. In brief, specific cells that express a nuclear mCherry are engineered to secrete a supercharged fluorescent protein (s36GFP) fused to the ultra-bright green-fluorescent mNeonGreen (mNG) (s36GFP). Neighboring cells uptake s36GFP and can be easily visualized. The authors added the human serum albumin signal peptide which is efficiently cleaved to create s36GFP. The PUFFFIN system can also be customized for color-of-choice labelling using HaloTags.

A shortcoming of the paper is that it is a method paper established in tissue culture cells with no biological applications. A test of the system in an in vivo model would improve the study. The authors should at least describe specific examples of how the method can be used to answer biological questions.

2. Significance:

Significance (Required)

This straightforward and elegant approach is an improvement of current methods that are based on synthetic receptor-ligand interactions as it does not require genetic modification of both 'sender' cells and 'responding' cells. The approach should prove to be an effective and flexible tool for illuminating cellular neighborhoods. An interesting potential application of the method is to effectively deliver proteins fused to s36GFP.

A shortcoming of the paper is that it is a method paper established in tissue culture cells with

no biological applications. A test of the system in an in vivo model would improve the study. The authors should at least describe specific examples of how the method can be used to answer biological questions.

3. How much time do you estimate the authors will need to complete the suggested revisions:

Estimated time to Complete Revisions (Required)

(Decision Recommendation)

Cannot tell / Not applicable

Yes

Review #3

1. Evidence, reproducibility and clarity:

Evidence, reproducibility and clarity (Required)

In this manuscript, the authors introduced a novel cell-neighbor-labeling system named PUFFFIN. PUFFFIN, as well as 'PUFFHalo', offers an elegantly simple method for distinguishing between secretors and receivers, providing users with a versatile tool to label proximate neighbors through the uptake of s36GFP, subsequently permitting their isolation via FACS for subsequent analysis. In addition, this system could be very useful considering of its customizability by exchanging elements, such as tissue-specific promoters, color-of-choice (HaloTag), and genes of interest to cater to the diverse requirements of secretors. Overall, this system is well-designed and characterized, and the claims in this study are mostly supported by the data. However, this neighbor-labeling approach is not efficiently used to obtain biological insights. The following comments are intended to enhance the overall quality of the study:

****Major comments:****

1. In Video1, it appears that certain nuclear mCherry+ cells did not secrete s36GFP-mNG during 19hrs recording window. However, in Figure1D and E, these GFP-mCherry+ cells were reported as having a 0% occurrence. This may be the result of either a delay in GFP secretion, or possible mCherry leakiness in unmodified cells. Please provide clarification. Additionally, including representative images of the co-culture experiment in Figure 1.E would enhance the presentation of the data.
2. Since the authors mention that s36GFP-mNG labeling was not detectable beyond four cell diameters, it would be helpful to include statistical data regarding the average distances or cell layers that GFP can travel, thus describing the permeation and labeling limit of s36GFP-mNG, adjacent to Figure2C.
3. Please comment on the application prospect of this system utilizing in vivo. In addition, comment should be made on the difference of PUFFFIN system and recent reported CILP (PNAS 2023).

****Minor comments:****

1. Please include the percentage of GFP+ and GFP- cells in Figure2.D, similar to what is provided in Figure S1.B.
2. The '+' and '-' marks in Figure3.E appears to be mismatched with the results, please double-check and correct.
3. I am curious about the interactions between secretors and 'receivers.' As the authors claim 'unbiased labeling' with this system, it's important to investigate whether the uptake abilities of receivers vary among different cell types. In other words, does the system exhibit cell-type preferences among receiver cells? This question could be optionally addressed through co-culture experiments involving secretors, receiver type A, and receiver type B.

2. Significance:

Significance (Required)

This study reported a simple and sensitive system for labeling neighboring cells in vitro, which can be customized by replacing exchangeable components for customized need. With promising application in vitro, this system could be further developed and tested in vivo. Fluorescent protein labeling in neighboring cells has been a topic of study recently, and this manuscript introduced a new tool that is added to such resources, offering a user-friendly and customizable alternative.

Overall, this system will be of interest to researchers working on neighbor-cell labeling and study of cell-cell communications.

3. How much time do you estimate the authors will need to complete the suggested revisions:

Estimated time to Complete Revisions (Required)

(Decision Recommendation)

Less than 1 month

Yes

Revision Plan

Manuscript number: RC- 2023-02169

Corresponding author(s): Sally Lowell sally.lowell@ed.ac.uk

1. **General Statements [optional]** please see cover letter

2. **Description of the planned revisions**

Reviewer #2 "A shortcoming of the paper is that it is a method paper established in tissue culture cells with no biological applications. A test of the system in an in vivo model would improve the study. The authors should at least describe specific examples of how the method can be used to answer biological questions."

Reviewer #3: "This neighbor-labeling approach is not efficiently used to obtain biological insights."

We are currently performing experiments to address a longstanding biological question and to test PUFFFIN in vivo. In a new advance since posting our preprint, we have established PUFFFIN in pluripotent cells and shown that it works beautifully in this developmentally-relevant context (Figure R1 below). This positions us to address the open question of how pluripotent cells communicate with their neighbours to generate diversity in the rate or direction of differentiation. We will provide these data in a new Figure 5 (please also see cover letter to Editor). In addition, we have three collaborators currently testing our system in developmental model organisms, and will include these data in our revised manuscript.

Figure for reviewers removed

Reviewer #3: minor comments

"I am curious about the interactions between secretors and 'receivers.' As the authors claim 'unbiased labeling' with this system, it's important to investigate whether the uptake abilities of receivers vary among different cell types. In other words, does the system exhibit cell-type preferences among receiver cells? This question could be optionally addressed through co-culture experiments involving secretors, receiver type A, and receiver type B."

We have new data demonstrating that PUFFIN labelling works very well in pluripotent cells (see Figure R1 above). We will perform additional experiments to address the reviewer's question by directly comparing labelling efficiency across different receiver cell-types. We will present these data in a new supplemental figure.

3. Description of the revisions that have already been incorporated in the transferred manuscript

Reviewer #1

1) The authors conclude that "PUFFIN labelling is transferred rapidly between cells within minutes". They report in their time lapse experiments (Figure 2A,C) that sGFP can be detected within neighbours of secretors after 30 minutes when the cells are plated in a 50:1 non-labelled/secretor cell ratio, whereas it can be detected after 15 minutes when the cells are plated in a 1:9 ratio. Is there any synergistic effect on the signal when the proportion of secretors is increased or is this difference just because there is more signal, making it easier to visualise.

We have addressed this point with new experiments (new data shown in Figure 2E and Supp Figure S2A,B). This makes it clear that labelling can indeed be detected earlier when the proportion of secretors is higher. This is likely to be because higher secretor:acceptor ratios result in stronger labelling, which in turn makes it easier to detect labelled neighbours at very early time points - even within as early as 15 minutes. We also confirm that, even when secretors are very sparse (1:50 ratio), label becomes detectable in neighbours within 60 minutes.

- 1. Is there any reason why the main Figure legends lack a title, but the supplementary figures have one?*
- 2. In Figure 3, it may be helpful to label each option as A, B, C..*
- 3. In Figure 4E, the legends + JF646 and -JF646 may be switched around. Shouldn't the orange box should be (+) and the grey box should be (-)?*

We thank the reviewer for these helpful points. We have modified / corrected the labelling as suggested and added titles to the main figure legends.

Reviewer #3

Revision Plan

1. In Video1, it appears that certain nuclear mCherry+ cells did not secrete s36GFP-mNG during 19hrs recording window. However, in Figure1D and E, these GFP-mCherry+ cells were reported as having a 0% occurrence. This may be the result of either a delay in GFP secretion, or possible mCherry leakiness in unmodified cells. Please provide clarification.

There is indeed one mCherry+ cell in video 1 that fails to generate s36GFP-mNG signal. This cell, unlike most other cells in the movie, fails to divide or actively migrate during the 19h recording period, but instead is being passively “pushed around” by surrounding cells, and therefore looks to us very much like a dead or dying cell (levels of cell death tend to be slightly higher than usual during live imaging). We have looked through our other videos and identified only one other example of an mCherry+ GFP-negative cell: this cell is clearly dying because the nucleus disintegrates over the course of the movie (Figure R2A)

We considered the possibility that some proportion of secretors may fail to generate signal even if they are healthy. We examined all our FACS analysis data (see examples on Figure R2B below). We detected at most 0.15% of such ‘failed secretors’, and usually far fewer or none. We conclude that any mCherry+ GFP- cells exist at extremely low frequencies and/or tend to be dying cells. Either way, they are very unlikely to interfere with interpretation of experimental data.

Figure for reviewers removed

Revision Plan

Additionally, including representative images of the co-culture experiment in Figure 1.E would enhance the presentation of the data.

Thank you for this suggestion. These data have now been added to Supplemental Figure S1 C

2. Since the authors mention that s36GFP-mNG labeling was not detectable beyond four cell diameters, it would be helpful to include statistical data regarding the average distances or cell layers that GFP can travel, thus describing the permeation and labeling limit of s36GFP-mNG, adjacent to Figure 2C.

We've now quantified the data and provide this information in a new panel (Figure 2D). This has considerably enhanced our analysis, so we are grateful to the reviewer for this excellent idea.

3. Please comment on the application prospect of this system utilizing in vivo. In addition, comment should be made on the difference of PUFFIN system and recent reported CILP (PNAS 2023).

We have added discussion on prospects for using the system in vivo (new text lines 65-67). We have also described the CILP system in the revised introduction, explaining that it is an inducible version of the Cherry Niche system that we describe in our introduction (new text lines 291-294).

Minor comments:

1. *Please include the percentage of GFP+ and GFP- cells in Figure 2.D, similar to what is provided in Figure S1.B.*

This is a great suggestion. We have decided to add this information to all flow cytometry histograms within the paper, not only Figure 2D.

2. *The '+' and '-' marks in Figure 3.E appears to be mismatched with the results, please double-check and correct.*

Thank you for spotting this error. This has now been corrected.

4. Description of analyses that authors prefer not to carry out

n/a.

Dear Prof. Lowell,

Thank you for transferring your manuscript with Review Commons referee reports and responses to The EMBO Journal.

Given the referees' positive recommendations, I would like to invite you to submit a revised version of the manuscript, addressing the comments of all three reviewers, particularly the inclusion of biological insight as suggested in your revision plan. I should add that it is EMBO Journal policy to allow only a single round of revision, and acceptance of your manuscript will therefore depend on the completeness of your responses in this revised version.

When preparing your letter of response to the referees' comments, please bear in mind that this will form part of the Review Process File, and will therefore be available online to the community. For more details on our Transparent Editorial Process, please visit our website.

Thank you for the opportunity to consider your work for publication. I look forward to your revision.

Yours sincerely,

Kelly M Anderson, PhD
Editor
The EMBO Journal
k.anderson@embojournal.org

We realize that it is difficult to revise to a specific deadline. In the interest of protecting the conceptual advance provided by the work, we recommend a revision within 3 months (4th Feb 2024). Please discuss the revision progress ahead of this time with the editor if you require more time to complete the revisions.

Rev_Com_number: RC-2023-02169

New_manu_number: EMBOJ-2023-116007

Corr_author: Lowell

Title: PUFFFIN: A novel, ultra-bright, customisable, single plasmid system for labelling cell neighbourhoods

EMBOJ-2023-116007

Lebek et al

PUFFFIN: A novel, ultra-bright, customisable, single plasmid system for labelling cell neighbourhoods

Response to reviewers

Reviewer #1 (Evidence, reproducibility and clarity):

This manuscript reports the development of a new bright fluorescent reporter that allows to label neighbouring cells. The system is based on upon secretion and uptake of s36GFP, a positively supercharged fluorescent protein. The authors also develop a Halo tag that will allow for straight forward colour exchange, as well as a customisable single plasmid construct with modular components.

There are some minor suggestions that the authors may want to consider:

1) The authors conclude that "PUFFIN labelling is transferred rapidly between cells within minutes". They report in their time lapse experiments (Figure 2A,C) that sGFP can be detected within neighbours of secretors after 30 minutes when the cells are plated in a 50:1 non-labelled/secretor cell ratio, whereas it can be detected after 15 minutes when the cells are plated in a 1:9 ratio. Is there any synergistic effect on the signal when the proportion of secretors is increased or is this difference just because there is more signal, making it easier to visualise.

1.1. We have addressed this point with new experiments (new data shown in Fig. 2E and Fig. EV2A,B). This makes it clear that labelling can indeed be detected earlier when the proportion of secretors is higher. This is likely to be because higher secretor:acceptor ratios result in stronger labelling, which in turn makes it easier to detect labelled neighbours at very early time points - even within as early as 15 minutes. These new data also confirm that, even when secretors are very sparse (1:50 ratio), label becomes detectable in neighbours within 60 minutes (Fig EV2A).

1. Is there any reason why the main Figure legends lack a title, but the supplementary figures have one?

2. In Figure 3, it may be helpful to label each option as A, B, C..

3. In Figure 4E, the legends + JF646 and -JF646 may be switched around. Shouldn't the orange box should be (+) and the grey box should be (-)?

1.2. We thank the reviewer for these helpful points. We have modified / corrected the labelling as suggested and added titles to the main figure legends.

Reviewer #2 (Evidence, reproducibility, and clarity)

The authors describe a new method, Positive Ultra-bright Fluorescent Fusion For Identifying Neighbours (PUFFFFIN), to label with Fluorescent Proteins, neighboring cells. In brief, specific cells that express a nuclear mCherry are engineered to secrete a supercharged fluorescent protein (36GFP) fused to the ultra-bright green-fluorescent mNeonGreen (mNG) (s36GFP). Neighboring cells uptake s36GFP and can be easily visualized. The authors added the human serum albumin signal peptide which is efficiently cleaved to create s36GFP. The PUFFFFIN system can also be customized for color-of-choice labelling using HaloTags.

A shortcoming of the paper is that it is a method paper established in tissue culture cells with no biological applications. A test of the system in an in vivo model would improve the study. The authors should at least describe specific examples of how the method can be used to answer biological questions.

2.1. We agree that a test of the system in an in vivo model would help demonstrate the utility of PUFFFFIN. We have now successfully tested the system in vivo and show these new data in Fig. 5 and Fig EV5. Our collaborators at the Crick delivered the PUFFFFIN system by electroporation into chick embryos and observed evidence of neighbour labelling by imaging (Fig. 5C) and by flow cytometry (Fig 5D). To further extend the utility of the PUFFFFIN system in vivo, we have shown that it can be packaged in a lentivirus and successfully delivered to rodent brain slices (Fig EV5).

2.2. We have also established the PUFFFFIN system in mouse pluripotent cells (Fig. 5E,F) and used it to obtain new biological insights into the longstanding question of whether

cells coordinate differentiation with neighbours (Fig. 6, Fig EV6). Until now it has been challenging to address this question because exit from naïve pluripotency is functionally defined using clonal commitment assays (Kalkan et al. 2017), limiting the utility of imaging-based approaches. Our PUFFFIN neighbour labelling system overcomes this challenge, allowing us to use functional assays to explore how pluripotent cells adjust the pace of differentiation to synchronise with their neighbours during exit from naïve pluripotency (Fig. 6, Fig EV6).

Reviewer #2 (Significance):

This straightforward and elegant approach is an improvement of current methods that are based on synthetic receptor-ligand interactions as it does not require genetic modification of both 'sender' cells and 'responding' cells. The approach should prove to be an effective and flexible tool for illuminating cellular neighborhoods. An interesting potential application of the method is to effectively deliver proteins fused to s36GFP.

A shortcoming of the paper is that it is a method paper established in tissue culture cells with no biological applications. A test of the system in an in vivo model would improve the study. The authors should at least describe specific examples of how the method can be used to answer biological questions.

2.3. Please see points 2.1 and 2.2 above, where we describe how we have established PUFFFIN labelling in vivo (Fig. 5 and Fig. EV5) and how we have used PUFFFIN to address a biological question (Fig. 6 and Fig. EV6).

Reviewer #3 (Evidence, reproducibility and clarity (Required)):

In this manuscript, the authors introduced a novel cell-neighbor-labeling system named PUFFFIN. PUFFFIN, as well as 'PUFFHalo', offers an elegantly simple method for distinguishing between secretors and receivers, providing users with a versatile tool to label proximate neighbors through the uptake of s36GFP, subsequently permitting their isolation via FACS for subsequent analysis. In addition, this system could be very useful considering of its customizability by exchanging elements, such as tissue-specific

promoters, color-of-choice (HaloTag), and genes of interest to cater to the diverse requirements of secretors. Overall, this system is well-designed and characterized, and the claims in this study are mostly supported by the data. However, this neighbor-labeling approach is not efficiently used to obtain biological insights. The following comments are intended to enhance the overall quality of the study:

Major comments:

1. In Vidio1, it appears that certain nuclear mCherry+ cells did not secrete s36GFP-mNG during 19hrs recording window. However, in Figure1D and E, these GFP-mCherry+ cells were reported as having a 0% occurrence. This may be the result of either a delay in GFP secretion, or possible mCherry leakiness in unmodified cells. Please provide clarification.

3.1a There is indeed one mCherry+ cell in video 1 that fails to generate s36GFP-mNG signal. This cell, unlike most other cells in the movie, fails to divide or actively migrate during the 19h recording period, but instead is being passively “pushed around” by surrounding cells, and therefore looks to us very much like a dead or dying cell (levels of cell death tend to be slightly higher than usual during live imaging).

We have looked through our other videos from this set of experiments and identified only one other example of an mCherry+ GFP-negative cell: this cell is clearly dying because the nucleus disintegrates over the course of the movie (see Fig. R1A below, provided for the reviewer to help us explain this point but not included in the manuscript) We considered the possibility that some proportion of secretors may fail to generate signal even if they are healthy. We examined all our FACS analysis data from this set of experiments. We detected at most 0.15% of such ‘failed secretors’, and most usually none. Some examples are provided in Fig R1B,C below, which is provided for the reviewer to help us explain this point but not included in the manuscript.

We conclude that any mCherry+ GFP- cells exist at extremely low frequencies in established PUFFFIN cell lines and/or that any such cells tend to be dying cells. Either way, they are very unlikely to interfere with interpretation of experimental data.

Furthermore, our new flow cytometry data from chick embryos (Fig. 5D) shows that expression s36-HaloTag PUFFFIN label tends to correlate well with expression of mCherry even in this in- vivo setting where expression is naturally variable (in contrast to the uniformly strong expression seen in established cell lines).

Figure for reviewers removed

Additionally, including representative images of the co-culture experiment in Figure 1.E would enhance the presentation of the data.

3.1b These data have now been added to Fig. EV1C

2. Since the authors mention that s36GFP-mNG labeling was not detectable beyond four cell diameters, it would be helpful to include statistical data regarding the average distances or cell layers that GFP can travel, thus describing the permeation and labeling limit of s36GFP-mNG, adjacent to Figure2C.

3.2 We've now quantified the data and provide this information in a new panel (Fig. 2D).

3. Please comment on the application prospect of this system utilizing in vivo. In addition, comment should be made on the difference of PUFFFIN system and recent reported CILP (PNAS 2023).

3.3 We provide new data (Fig. 5A-D) showing that the PUFFFIN system can be used in chick embryos and also provide data showing that the PUFFFIN system can be delivered to mouse brain organotypic slice cultures (please see response 2.1 to reviewer #2 for more details). We have added discussion on broader prospects for using the system in vivo (lines 382 - 387). We have also described the CILP system in the revised introduction, explaining that it is an inducible version of the Cherry Niche system that we describe in our introduction (line 69).

Minor comments:

1. Please include the percentage of GFP+ and GFP- cells in Figure2.D, similar to what is provided in Figure S1.B.

3.4 This is a great suggestion so we have decided to add this information to all flow cytometry histograms within the paper.

2. The '+' and '-' marks in Figure3.E appears to be mismatched with the results, please double-check and correct.

3.5 Thank you. This has now been corrected.

3. I am curious about the interactions between secretors and 'receivers.' As the authors claim 'unbiased labeling' with this system, it's important to investigate whether the uptake abilities of receivers vary among different cell types. In other words, does the system exhibit cell-type preferences among receiver cells? This question could be

optionally addressed through co-culture experiments involving secretors, receiver type A, and receiver type B.

3.6: We provide new data (Fig. EV1D) showing that there is no obvious difference in labelling efficiency across several different receiver cell-types.

Reviewer #3 (Significance (Required)):

This study reported a simple and sensitive system for labeling neighboring cells in vitro, which can be customized by replacing exchangeable components for customized need. With promising application in vitro, this system could be further developed and tested in vivo. Fluorescent protein labeling in neighboring cells has been a topic of study recently, and this manuscript introduced a new tool that is added to such resources, offering a user-friendly and customizable alternative.

Overall, this system will be of interest to researchers working on neighbor-cell labeling and study of cell-cell communications.

We thank all three reviewers for their insightful comments and suggestions.

Reference cited within this response to reviewers

Kalkan T, Olova N, Roode M, Mulas C, Lee HJ, Nett I, Marks H, Walker R, Stunnenberg HG, Lilley KS, Nichols J, Reik W, Bertone P, Smith A. Tracking the embryonic stem cell transition from ground state pluripotency. *Development*. 2017 Apr 1;144(7):1221-1234. doi: 10.1242/dev.142711.

Dear Sally,

Congratulations on a great revision! Overall, the referees have been positive. However there remain a few editorial items that we ask you to complete in a revised version. When you submit your revised version, please address the following and add to your point-by-point response:

1. Please add the University of Edinburgh School of Biological Sciences, Wellcome Centre for Cell Biology grant 203149 online to eJP. If necessary, please also include the University of Edinburgh Chancellor's Fellowship and the Simons Initiative for the Developing Brain in the list of funders.
2. Please move the key words to follow the abstract.
3. Please add a callout to the movie files in the main manuscript.
4. We include a synopsis of the paper (see <http://emboj.embopress.org/>). Please provide me with a general summary statement and 3-5 bullet points that capture the key findings of the paper.
5. We also need a summary figure for the synopsis. The size should be 550 wide by 200-440 high (pixels). You can also use something from the figures if that is easier.
6. Please rename the Structured Methods to Methods.
7. Please remove the reagent table from the manuscript file.
8. Please remove the movie legends from the main manuscript. Each legend should be provided in a readme.txt file and the should be zipped together with its corresponding movie so that we have folder per movie uploaded.
9. Please define the scale bar for figure 5f.
10. Please add a scale bar and its definition for figure EV 4d(1-2).

Thank you for the opportunity to consider your work for publication. I look forward to your revision.

Warm regards,
Kelly

Kelly M Anderson, PhD
Editor, The EMBO Journal
k.anderson@embojournal.org

Referee #1:

The authors have addressed my comments and added some in vivo data. The manuscript is acceptable for publication.

Referee #2:

The authors have addressed the two minor points made in the original review. This is a very elegant tool and a very nice article. The new functional data obtained using this tool highlights how useful PUFFIN will be. It also gives very valuable insight into the onset of differentiation.

Referee #3:

The authors have addressed my questions adequately during the revision. The only limit that remains is that there is no genetic tool generated from PUFFFIN for use of in vivo biological study.

Rev_Com_number: RC-2023-02169

New_manu_number: EMBOJ-2023-116007R

Corr_author: Lowell

Title: PUFFFIN: A novel, ultra-bright, customisable, single plasmid system for labelling cell neighbourhoods

Response to reviewers

We respond to all three reviewers at the end of their combined comments

Referee #1: The authors have addressed my comments and added some in vivo data. The manuscript is acceptable for publication.

Referee #2: The authors have addressed the two minor points made in the original review. This is a very elegant tool and a very nice article. The new functional data obtained using this tool highlights how useful PUFFIN will be. It also gives very valuable insight into the onset of differentiation.

Referee #3: The authors have addressed my questions adequately during the revision. The only limit that remains is that there is no genetic tool generated from PUFFIN for use of in vivo biological study.

We thank all three reviewers for their positive comments.

We agree with Referee #3 that it would be interesting to explore genetic tools, for example by generating mouse lines that are engineered to express PUFFIN in particular cell types or disease states, but this would be a lengthy and expensive process and we feel it is beyond the scope of this current manuscript. We think it likely that many labs will be able to perform in-vivo neighbour-labelling experiments quickly and cheaply using the electroporation or viral delivery methods we report in our manuscript.

Dear Sally,

Congratulations on an excellent manuscript, I am pleased to inform you that your manuscript has been accepted for publication in The EMBO Journal. Thank you for your comprehensive response to the referee concerns and for providing detailed source data. It has been a pleasure to work with you to get this to the acceptance stage.

I will begin the final checks on your manuscript before submitting to the publisher next week. Once at the publisher, it will take about 3 weeks for your manuscript to be published online. As a reminder, the entire review process, including the referee concerns and your point-by-point response, will be available to readers.

I will be in touch throughout the final editorial process until publication. In the meantime, I hope you find time to celebrate!

Warm wishes,
Kelly

Kelly M Anderson, PhD
Editor, The EMBO Journal
k.anderson@embojournal.org
